# Memristor-based adaptive analog-to-digital conversion for efficient and accurate compute-in-memory

Haiqiao Hong [1], Zhiyuan Du[1], Mingrui Jiang [1], Ruibin Mao [1], Yuan Ren [1], Fuyi Li[2], Wei Mao[2], Muyuan Peng[1], Wei Zhang[3], Zhengwu Liu [1] ✉, Can Li [1] ✉ & Ngai Wong [1] ✉

Compute-in-memory technology offers promising solutions for neural network acceleration but its potential is severely limited by inflexible and resource-intensive analog-to-digital converters. Here, we present a memristor-based analog-to-digital converter featuring adaptive quantization for diverse output distributions. Our design employs analog content-addressable memory cells with programmable overlapped boundaries to establish optimized quantization thresholds, demonstrating excellent integral and differential non-linearities. Extensive experiments validate the robustness of our approach by achieving 89.55% accuracy on CIFAR-10 (VGG8) at 5-bit adaptive quantized precision and maintaining competitive performance on ImageNet (ResNet18) through a proposed super-resolution strategy under experimental memristor variations. Compared to state-of-the-art designs, our converter achieves a 15.1× improvement in energy efficiency and a 12.9× reduction in area. Furthermore, integrating our converter into CIM systems reduces the energy and area overhead by up to 57.2% and 30.7%, respectively. This work establishes a paradigm for efficient and accurate signal quantization in practical compute-in-memory systems.

As artificial intelligence (AI) advances rapidly, the increasing complexity of deep neural networks exposes critical limitations in traditional von Neumann architectures[1,2]. These architectures, with their separate computation and memory units[3], suffer from massive data movement overheads in modern neural network operations[4,5], resulting in high energy consumption and latency[6–8]. Compute-in-memory (CIM) with emerging memory devices, particularly memristors, offers a promising solution by integrating computation directly within memory crossbar arrays[9–19] (Fig. 1a). This integration overcomes the von Neumann bottleneck and enables efficient vector-matrix multiplications (VMMs) for deep learning operations[15,20–23]. A typical mixed-signal CIM system (Fig. 1b) consists of digital-to-analog converters (DACs), analog crossbar arrays, analog-to-digital converters (ADCs),

and digital activation circuit implementing linear operations, quantization, and nonlinear activation functions in neural network layers, respectively.

Despite these advantages, analog-to-digital signal conversion emerges as a critical bottleneck in CIM systems[10,24–27]. Current ADCs, as fundamental components for signal conversion, consume excessive energy and circuit area, fundamentally constraining the scalability of CIM systems as neural network complexity grows. For example, in a state-of-the-art implementation[28], ADCs occupy up to 87.8% of total energy and 75.2% of chip area (Fig. 1c). Moreover, existing quantization methods struggle to handle diverse output distributions in CIM architectures[29,30] in an energy-efficient manner. In CIM implementations, different neural network layers and even channels are mapped to

[1]Department of Electrical and Electronic Engineering, The University of Hong Kong, Hong Kong, China. [2]Hangzhou Institute of Technology, Xidian University, Hangzhou, China. [3]Department of Electronic and Computer Engineering, The Hong Kong University of Science and Technology, Hong Kong, China. ✉e-mail: zwliu@eee.hku.hk; canl@hku.hk; nwong@eee.hku.hk

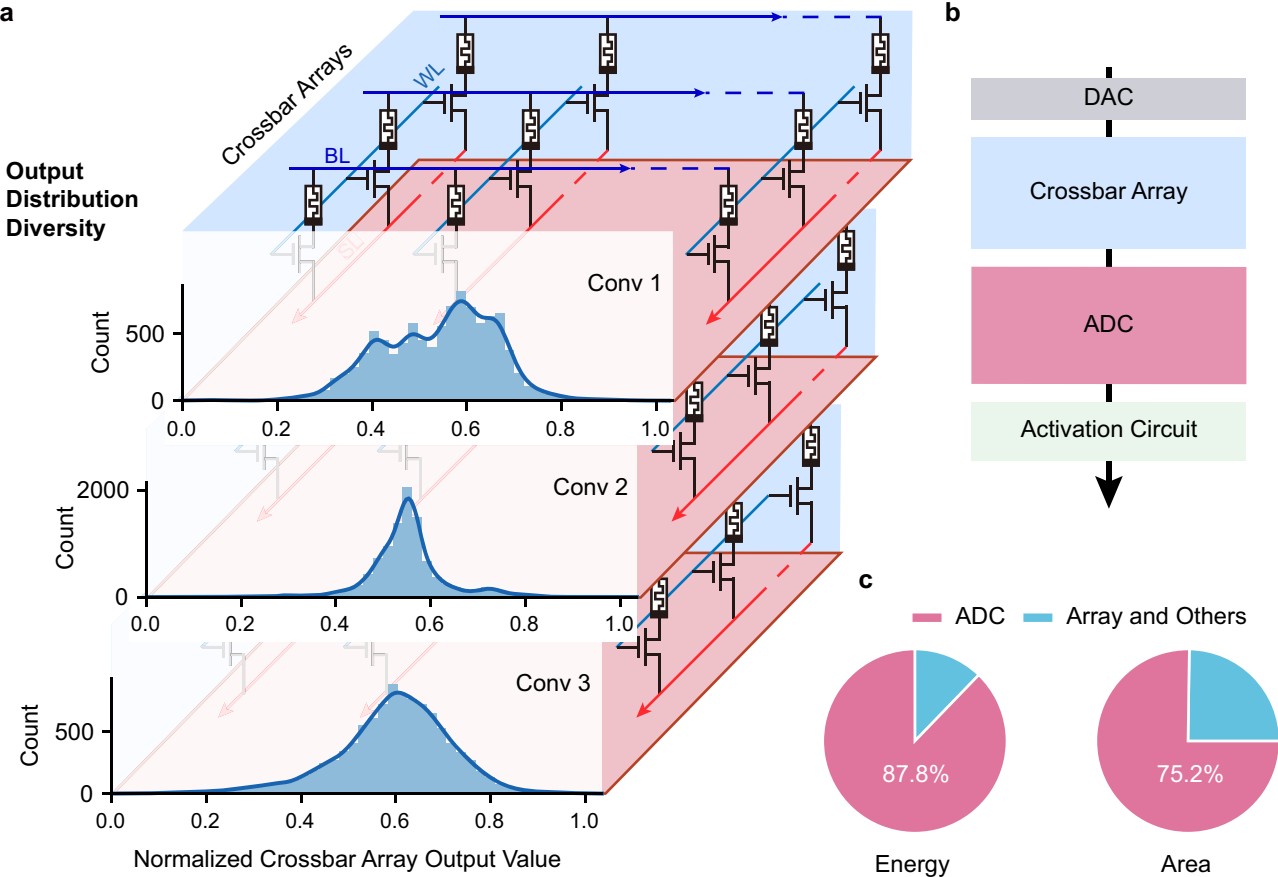

**Fig. 1 | Challenges in CIM systems for neural network computation. a** Output distributions from simulated crossbar arrays for three convolutional layers in VGG8 show significant diversity across layers. This diversity presents challenges for precision-constrained uniform quantization. **b** Schematic of a typical mixed-signal CIM system, showing the key components involved in neural network computation. The crossbar array executes VMMs, followed by ADC converting analog outputs into digital signals for subsequent digital nonlinear activation functions. **c** Resource breakdown in a state-of-the-art CIM implementation illustrates the substantial portion of energy and area consumed by ADCs.

separate crossbar arrays, each carrying distinct weight patterns. This mapping, combined with varying input patterns during inference, leads to significantly different output distributions across arrays (Fig. 1a), demanding quantization schemes that are both adaptive and efficient. Current solutions present a clear trade-off: at low ADC precision (≤6 bits), uniform quantization generally offers hardware-friendly implementation with low energy and area overhead but suffers from significant precision loss[2], while adaptive approaches based on cumulative distribution function[31] and Lloyd-Max algorithm[32–34] achieve higher accuracy through dynamic boundary adjustment but lack practical hardware implementations. At higher ADC precision, uniform quantization provides sufficient accuracy, making adaptive quantization less advantageous. However, the significant energy and area overhead of high-precision ADCs substantially diminishes the efficiency benefits of CIM systems, making low-bit implementations crucial for maintaining CIM's advantages in hardware efficiency.

To address these fundamental challenges, we introduce a memristor-based ADC that leverages the intrinsic programmability of memristors to enable adaptive quantization directly in hardware. It uses analog content-addressable memory (CAM) cells to implement this approach in CIM systems, significantly reducing quantization errors compared to uniform approaches. By establishing quantization thresholds with overlapped CAM boundaries, our design demonstrates excellent quantization performance with integral non-linearity (INL) and differential non-linearity (DNL) of 0.319 and 0.419 least significant bit (LSB) for a 5-bit configuration. In neural network

applications, our ADC exhibits robust performance across different architectures under practical memristor variations. The 5-bit ADC achieves 89.55% accuracy on CIFAR-10 using VGG8. For more complex networks like ResNet18 on ImageNet, our super-resolution strategy effectively mitigates hardware variations, achieving near-ideal accuracies of 34.76% (4-bit), 61.81% (5-bit), and 65.50% (6-bit), respectively. Moreover, the ADC exhibits high hardware efficiency, delivering a 15.1× lower energy consumption while reducing the chip area by 12.9× compared to state-of-the-art solutions. Furthermore, compared to memristor ramp-based ADC implementation[35], our one-step conversion ADC achieves 48.5× lower latency and 26.6× lower energy consumption while maintaining comparable area efficiency. When integrated into CIM systems and compared to conventional successive approximation register (SAR) ADC, our design significantly reduces ADC overhead, decreasing energy and area contribution by 57.2% and 30.7% respectively for VGG8, and by 56.9% and 25.1% for ResNet18.

## Results

### Design of the memristor-based ADC
Our ADC design leverages analog CAM[36–38] as the core to create a programmable voltage quantizer, termed a quantization cell (Q-cell) (Fig. 2a). The Q-cell consists of two memristors ($M_1$, $M_2$), five transistors ($T_1$ to $T_5$), and two inverters ($INV_1$, $INV_2$). $V_H$ and $V_L$ are applied across voltage dividers formed by $M_1$-$T_1$ and $M_2$-$T_3$ pairs, while restricted to avoid mis-programming the device conductance. By tuning the memristor conductance, quantization boundaries can be

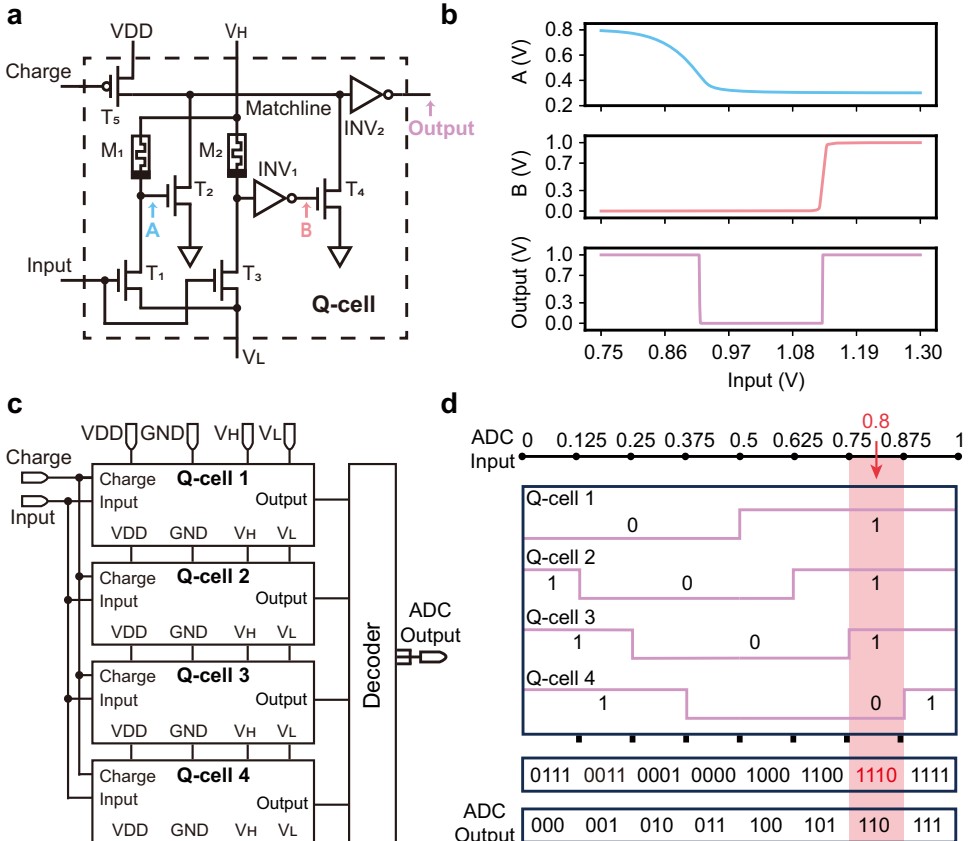

**Fig. 2 | Memristor-based ADC design. a** Schematic of a memristor-based Q-cell. **b** Simulated responses to the input voltage, including voltages at nodes A and B, and the output of the Q-cell. **c** Schematic of the proposed ADC at 3-bit precision, showing the arrangement of four Q-cells and decoder to achieve 3-bit precision quantization. **d** Illustration of the proposed ADC at 3-bit precision. When input 0.8 V is applied, the cell output is 1110, and the ADC output is decoded to 110.

defined through these dividers. Specifically, the $M_1$-$T_1$ divider controls the lower boundary via discharge transistor $T_2$, while the $M_2$-$T_3$ divider, coupled with $INV_1$, determines the upper boundary through discharge transistor $T_4$. This dual-boundary configuration enables voltage quantization based on programmed thresholds, as verified through circuit simulations using the UMC 180 nm technology library. The voltage responses at nodes A and B were obtained with a sampling time of 2 μs, with $V_H$ and $V_L$ set to 0.8 V and 0.3 V, respectively, across the $M_1$-$T_1$ and $M_2$-$T_3$ pairs. The input voltage was swept from 0.75 V to 1.30 V to characterize the Q-cell's response at nodes A and B (Fig. 2b). A PMOS transistor ($T_5$) initializes the circuit by charging the match line. The match line voltage, after passing through $INV_2$, generates a binary output: '0' when the input voltage falls within the programmed boundaries, and '1' otherwise.

To illustrate the fundamental operation of our ADC architecture, we first present its implementation with uniform quantization, which allows us to validate the basic functionality and reliability of our design through well-established ADC metrics. This uniform configuration serves as a foundation for understanding the device's capabilities before exploring its adaptive quantization features.

The ADC precision determines the required number of Q-cells, with an n-bit ADC requiring $2^{n-1}$ Q-cells to cover the input range (see Supplementary Note 1 for details). Figure 2c, d shows a 3-bit ADC implementation using four Q-cells, where seven quantization boundaries divide the normalized input range into eight uniformly distributed segments. Each Q-cell employs a progressive boundary distribution pattern based on LSB, with unique lower and upper boundary pairs. To maximize conductance range utilization, we implement a dual-bias strategy: the left input ($T_1$) receives a bias

voltage matching the minimum input range of the ADC, while the right input receives a smaller bias (half of the input range). This approach ensures balanced conductance utilization across both sides, enabling the same conductance range to define both lower and upper quantization boundaries. In our 3-bit implementation, the Q-cells are programmed with voltage ranges of (0, 0.5), (0.125, 0.625), (0.25, 0.75), and (0.375, 0.875). This overlapping configuration generates unique digital codes through distinct Q-cell activation patterns. For example, an input of 0.8 V activates only the last Q-cell (falling within 0.75–0.875), producing an output code of '1110', which is then decoded to the final binary output '110' (see Supplementary Note 2 and Supplementary Table 1 for details).

Although the illustrative example shows uniform quantization, the quantization boundary in our design is reconfigurable to achieve adaptive quantization. The key advantage of our design lies in its hardware efficiency and adaptability, where memristor programmability enables dynamic boundary reconfiguration to accommodate varying system requirements[39].

## Experimental benchmarking of uniform quantization ADC

The experimental characterization of our memristor-based ADC architecture necessitates a rigorous evaluation framework that encompasses both conventional ADC performance metrics and adaptive quantization capabilities. Given that the fundamental operation of our design relies on precise analog-to-digital conversion, we first evaluated its baseline performance using metrics such as INL and DNL under uniform quantization conditions. These metrics provide quantitative validation of the device's intrinsic conversion accuracy and linearity, which underpin both conventional and adaptive quantization functionalities.

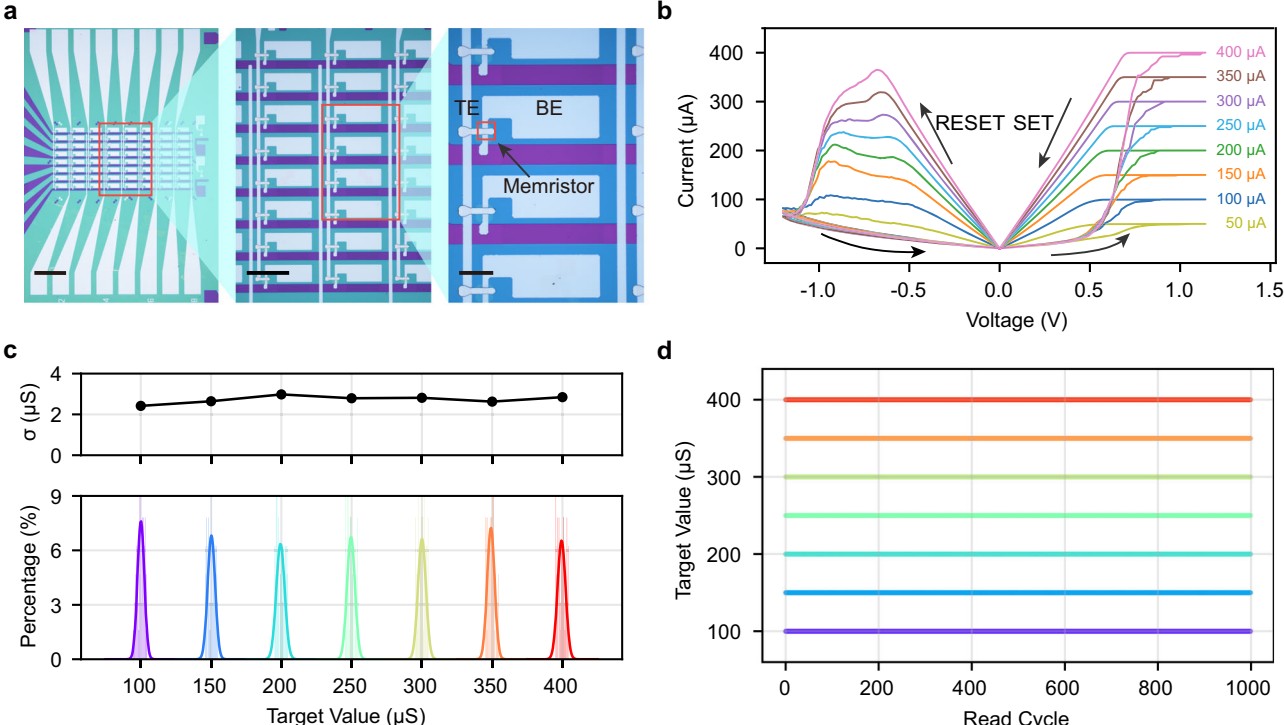

**Fig. 3 | Experimental characterization of the fabricated memristor device and array. a** Microscopy images of the fabricated 8×8 memristor crossbar array (left, middle) and a magnified view of an individual cross-point cell (right), showing the device structure with top (TE) and bottom (BE) electrodes. The scale bars are 300, 100, and 30 μm from left to right, respectively. **b** SET and RESET curves of a representative device, demonstrating that multiple distinct memristor conductance levels can be reliably realized by controlling the programming compliance current. **c** Experimental device-to-device variation measured across all 64 devices of the 8 × 8 array. The figure presents the conductance distributions for seven target states (100, 150, 200, 250, 300, 350, and 400 μS), alongside the corresponding standard deviation (σ) for each state, confirming consistent programming with σ values around 2.73 μS. **d** Read cycle-to-cycle variation characterization of a representative device, showing stable conductance (SD: 0.14 μS, 0.047%) over 1000 cycles for multiple programmed states.

To experimentally demonstrate our ADC design, we fabricated and characterized an 8×8 memristor array comprising 64 devices (Fig. 3a). The devices exhibit stable, multi-level conductance states controlled by programming compliance current (Fig. 3b), which is fundamental for establishing quantization boundaries. Statistical analysis of the array revealed device-to-device variation with a standard deviation of 2.73 μS for programmed states in the range of 100–400 μS, corresponding to <1% relative variation (Fig. 3c). Excellent read stability was demonstrated with only 0.14 μS (0.047%) conductance variation over 1000 consecutive cycles (Fig. 3d). These experimental characterizations (including Supplementary Figs. 1–3) validate the variation parameters to be used in our following simulation evaluations. Furthermore, endurance testing confirmed robust switching over $3 \times 10^7$ cycles within the required conductance window (Supplementary Fig. 4). For ADC applications where write operations are infrequent, this endurance projects to a device lifetime of ~87 years under a conservative estimate of 1000 daily updates, ensuring excellent long-term reliability.

Further analysis of device variability under programming conditions[40] extended our static characterization to encompass both INL and DNL[41] across the full operating range (see Methods for details). INL quantifies deviation from ideal linear response, while DNL measures uniformity between successive codes. In CIM systems, INL affects feature map computations, where high values lead to reduced inference accuracy, while DNL determines precision in distinguishing activation levels. The impact of these variations is amplified in CIM architectures, as ADC quantization across array lines can cascade through network layers, causing significant accuracy loss in deep learning tasks.

For our 4-bit ADC implementation, maximum INL and DNL were limited to 0.215 LSB and 0.322 LSB, respectively (Fig. 4a, b), validating

design robustness under experimental variation. Scalability analysis across 2-6 bit resolutions (Fig. 4c, d) using root mean square (RMS) analysis of maximum INL and DNL revealed consistent performance (see Methods for details). As detailed in Supplementary Table 2, all bit precisions maintained sub-1-LSB linearity metrics under experimental variation. Specifically, at 5-bit precision, our architecture achieved RMS values of 0.319 LSB and 0.419 LSB for maximum INL and DNL, respectively. Beyond device conductance variations, the ADC also demonstrates strong resilience to dynamic power supply noise and offers an in-situ recalibration mechanism to counteract static DC drifts, as detailed in Supplementary Note 3 and Supplementary Fig. 5. These measurements, conducted under uniform quantization, establish the fundamental precision and stability of our ADC architecture, providing critical attributes that enable subsequent implementation of adaptive quantization schemes across diverse CIM applications.

## Adaptive quantization ADC for efficient neural networks

Although our ADC design has demonstrated competitive benchmark performance, its full potential has yet to be exploited. What makes it special is that the quantization boundaries can be programmed arbitrarily using the memristors in the Q-cell, opening up the possibility for adaptive quantization in neural networks and achieving non-degraded performance even with low-bit quantization. Our adaptive quantization implementation integrates hardware-based boundary adjustment with software optimization. This process encompasses both offline analysis and dynamic operation during inference. Initially, we collect output statistics from each convolution layer by running inference on a representative training dataset, capturing the distinct characteristics of various network layers. Utilizing these statistics, we apply a modified Lloyd-Max algorithm[32] to determine optimal quantization boundaries

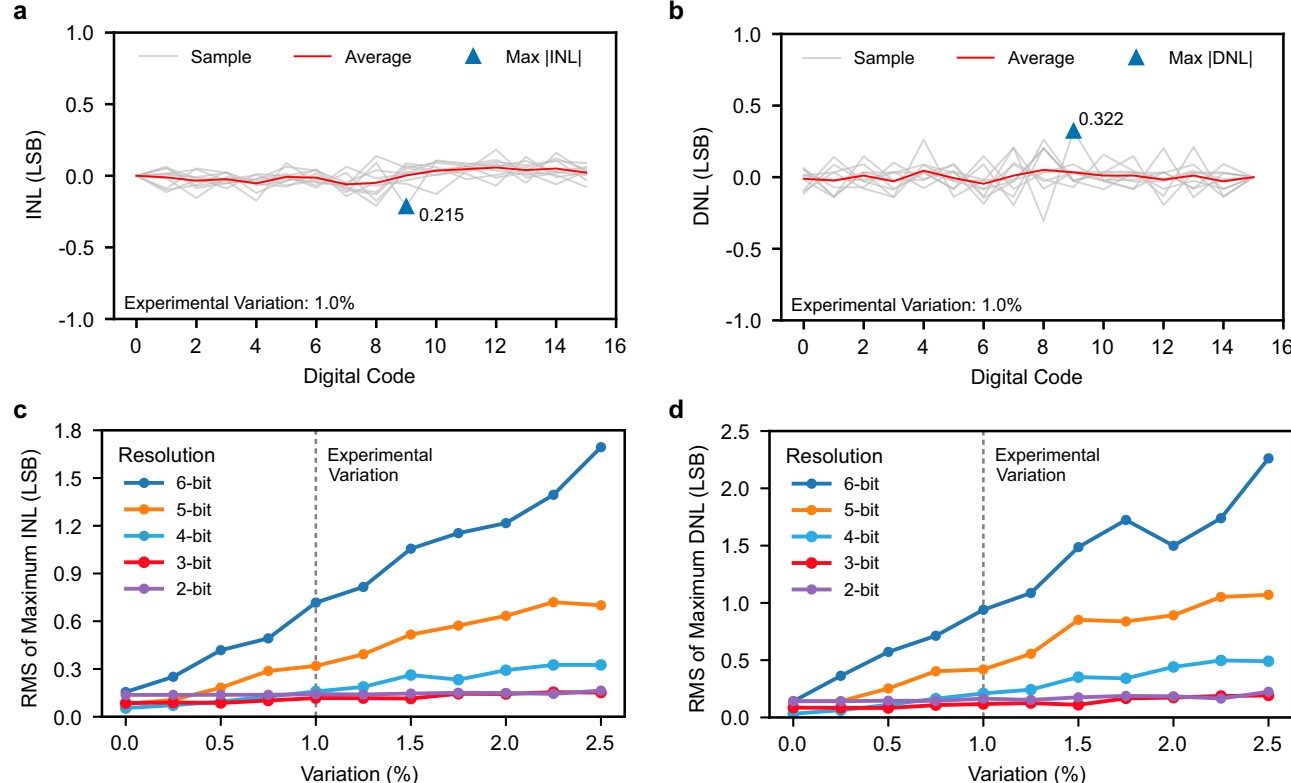

**Fig. 4 | Simulated performance metrics of the memristor-based ADC. a** INL of 4-bit ADC under 1% variation across 16 digital codes, maximum 0.215 LSB. **b** DNL under the same conditions, maximum 0.322 LSB. RMS analysis of maximum INL (**c**) and DNL (**d**) under various device variations for ADC resolutions from 2 to 6 bits, illustrating the ADC's sensitivity to such variations.

(Supplementary Fig. 6), iteratively adjusting them to minimize mean squared error (MSE) while adhering to hardware constraints. Subsequently, we program these optimized boundaries into the memristor devices through precise conductance modulation, employing either compliance current control or programming pulses with program-and-verify calibration. This approach accounts for the non-linear relationship between conductance and voltage thresholds in our Q-cell design. During inference, each Q-cell operates with these optimized boundaries, enabling adaptive quantization that closely aligns with the actual distribution of CIM array outputs.

This software-hardware co-optimization methodology ensures the effective realization of adaptive quantization's theoretical benefits in physical implementation. The hardware component exploits memristors' programmable conductance states for dynamic quantization boundary setting, while the software optimization process fine-tunes these boundaries based on actual data distributions and hardware constraints, resulting in a highly efficient and adaptable CIM system.

To validate our approach, we evaluated our strategy using the VGG8 network[42] on the CIFAR-10 dataset[43]. The experimental results demonstrated the effectiveness of our adaptive quantization method. Specifically, the reconfigurable ADC achieved improved boundary alignment with data distribution compared to uniform quantization (Fig. 5a), reducing MSE from 14.99 to 3.10 (Fig. 5b).

Across different ADC precisions (Fig. 5c), our adaptive approach achieved 68.1%, 88.9%, and 90.2% accuracy at 3-bit, 4-bit, and 5-bit, respectively, significantly outperforming uniform quantization (13.0%, 52.3%, 89.9%). Under experimental variations (Fig. 5d), the network maintained robust performance with 66.30%, 88.39%, and 89.55% accuracy at respective bit levels.

For the more complex ResNet18 network[44] on ImageNet[45], we conducted a comprehensive evaluation across different quantization scenarios. First, we established baseline performance using ideal

adaptive quantization, achieving 30.7%, 61.4%, and 67.5% accuracy at 4-bit, 5-bit, and 6-bit precisions, respectively. These results exhibit the theoretical benefits of adaptive quantization at different bit precisions (Fig. 5e). However, when implementing the ADC with real memristor devices under experimental variations (1% conductance variation), the accuracies dropped to 24.1%, 51.3%, and 59.0%. This accuracy degradation (average 6.6% drop) reveals the impact of device variations on quantization precision.

To address this challenge, we developed a super-resolution strategy that leverages the inherent variation between paired ADCs. When two ADCs with nominally identical boundaries produce different outputs due to device variations, it suggests that the input lies within the boundary region affected by these variations. We identify such boundary-proximate states, assign appropriate boundary values, and thereby utilize device variations for super-resolution (see Methods for implementation details). This approach effectively improved accuracies to 34.8%, 61.8%, and 65.5% for 4-bit, 5-bit, and 6-bit configurations respectively (Fig. 5f). Notably, these results not only recover the accuracy loss from device variations but, in some cases (e.g., 4-bit: +4.06% improvement), also surpass the theoretical adaptive quantization baseline. This improvement shows how our super-resolution strategy transforms device variations from a limitation into an advantage, thereby enhancing quantization precision.

The above results also reveal that the performance advantage of our adaptive quantization scales with network complexity. For simpler networks like VGG8, benefits are most pronounced at very low bit-widths (+55.1% at 3-bit), while the accuracy gap narrows at 5-bit as uniform quantization becomes adequate. Conversely, complex networks like ResNet18 on ImageNet generate more diverse output distributions, extending the value of adaptivity to higher bitwidths. Rather than exhibiting diminishing returns, our approach delivers a substantial +17.3% absolute accuracy improvement at 6-bit (65.5% with

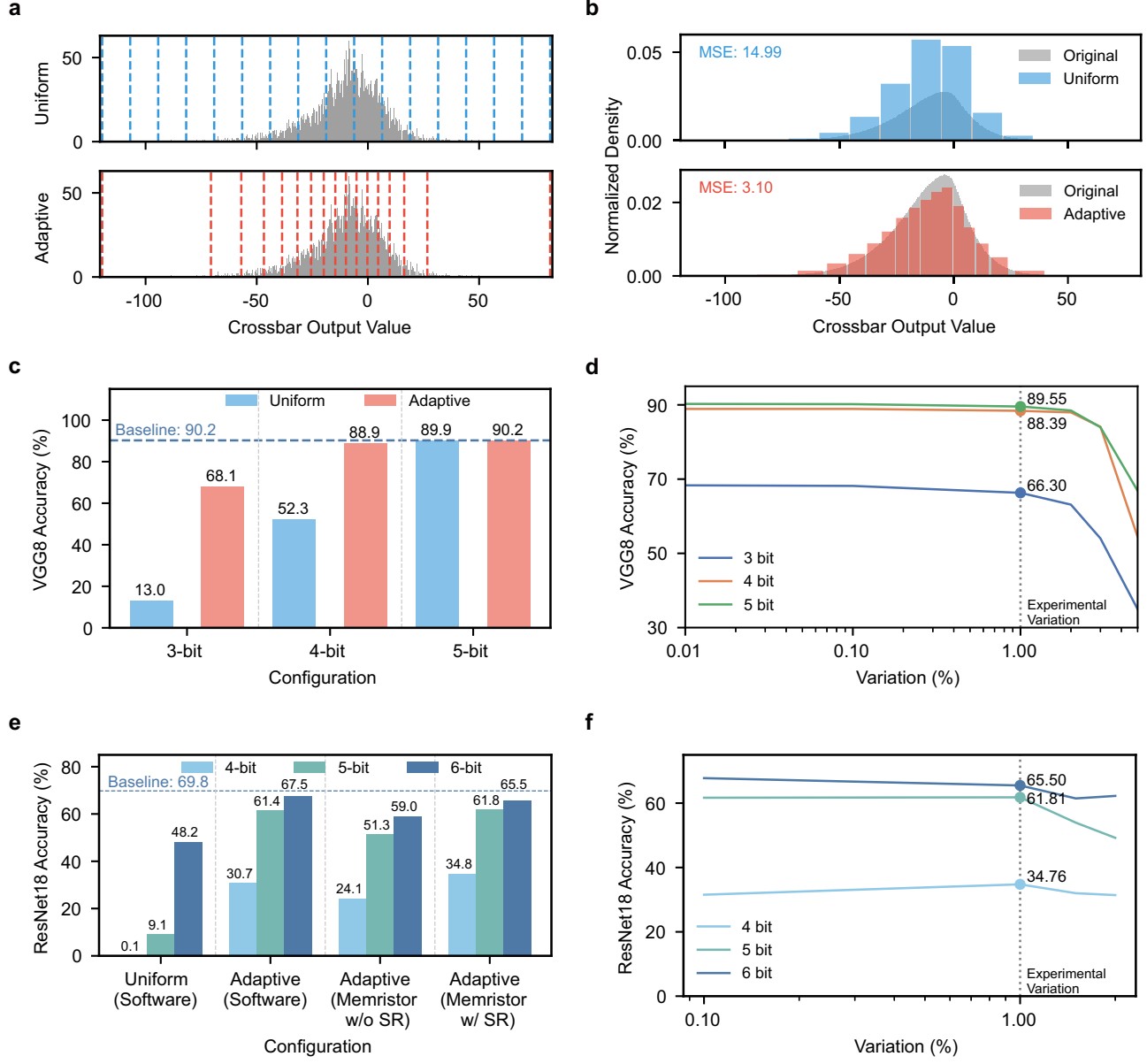

**Fig. 5 | Simulation analysis of adaptive quantization and network performance with our ADC. a** Distribution of crossbar array output values showing adaptive *vs.* uniform quantization boundaries. **b** MSE comparison between adaptive (3.10) and uniform (14.99) quantization on the full dataset. **c** VGG8 performance comparison: uniform *vs.* adaptive quantization at 3–5 bit precisions. **d** VGG8 accuracy across ADC precisions. **e** ResNet18 accuracy comparison: uniform quantization (software), adaptive quantization (software), our ADC without and with super-resolution (SR) strategy (under 1% experimental memristor variation) at 4–6 bit precisions. **f** ResNet18 accuracy with super-resolution strategy at 4–6 bit precisions.

SR vs. 48.2% for uniform), demonstrating effectiveness well beyond the "ultra-low bit" regime.

### Hardware efficiency evaluation of memristor-based ADC

To evaluate the hardware benefits of our approach, we conducted a comprehensive analysis comparing our ADC design with state-of-the-art SAR, Flash, and other ADCs from a public survey[46] of designs presented at the ISSCC and VLSI symposia between 1997 and 2024. Given the emerging nature of low-bit ADCs at advanced technology nodes, direct comparisons at 16 nm node with 5–6 bit precision are challenging due to the limited number of published works in this specific domain. To provide a comprehensive evaluation of our design's capabilities, we extended our comparison across different technology nodes and precision levels, enabling a broader assessment of the advantages offered by our memristor-based approach.

As illustrated in Fig. 6a, our memristor-based ADC demonstrates significant improvements in both energy efficiency and area utilization at 5-bit ADC precision across this comprehensive comparison space. A detailed breakdown of both energy consumption and area utilization (Fig. 6b) reveals the relative contributions of the ADC core components and the decoder, demonstrating the efficient resource allocation in our design. The design achieves an energy efficiency of 12.58 fJ per conversion (see Methods for details), representing a 15.1× improvement over conventional designs[47]. In terms of area, our ADC occupies only 24.29 µm², which is 12.9× smaller than the most compact conventional ADCs[48]. We also designed a full custom layout of the proposed ADC design (Supplementary Fig. 7). Besides, we conducted a comprehensive analysis comparing our ADC with a detailed comparison against CIM-specific designs provided in Supplementary Table 3.

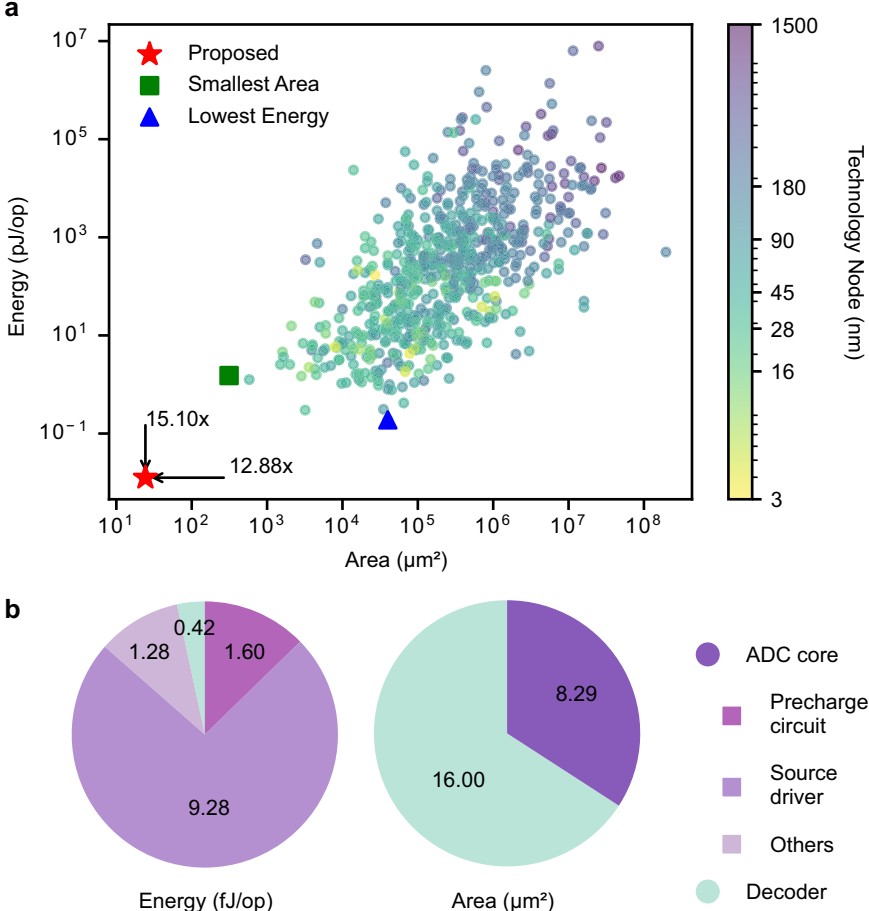

**Fig. 6 | Estimated energy and area comparison of memristor-based ADC.**
**a** Scatter plot comparing energy efficiency versus area of the state-of-the-art
ADCs[46], with technology nodes indicated by the color bar. Our memristor-based
ADC (red star) demonstrates competitive performance in both metrics, compared
to conventional designs, including lowest-energy design[47] (blue triangle) and
smallest-area design[48] (green square). **b** Breakdown of energy consumption and
area utilization of our ADC implemented at 5-bit precision, showing the proportion
between ADC core (precharge circuit, source driver, and other circuits) and
decoder.

Compared to existing memristor ramp-based ADC imple-
mentations[35], our one-step conversion mechanism demonstrates
advantages in both speed and energy efficiency (Supplementary
Table 4). Ramp-based ADCs achieve complete parallelism by allocating
a dedicated comparator to each column, enabling simultaneous
quantization of all column outputs. This architecture is exceptionally
well-suited for CIM systems that require high readout bandwidth.
Consequently, these systems become optimal for data-intensive and
high-throughput applications where the simultaneous processing of
large information volumes is critical. In scenarios requiring lower
latency, the architecture may be less optimal due to its operational
principle, which requires sequential comparisons as the ramp signal
progresses through each quantization. The prolonged comparator
activation during this comparison period leads to elevated energy
dissipation. For example, this principle results in a reported latency of
32 ns and energy consumption of 42.82 pJ at 5-bit precision. In con-
trast, our design achieves quantization in a single step, with all
thresholds evaluated simultaneously. Our implementation demon-
strates a latency of 0.66 ns and energy consumption of 1.61 pJ per VMM
computation in a configuration of 32 ADCs per macro, while main-
taining area efficiency (777.28 μm²). These improvements represent a
48.5× reduction in conversion latency and a 26.6× reduction in energy
consumption.

While our single-stage adaptive memristor-based ADC excels in
latency and energy for low-to-mid bitwidths, its flash-like architecture
faces a scalability challenge for achieving very high resolutions due to
the exponential growth of Q-cells. For such applications, a promising
direction is to employ a multi-stage pipeline architecture, where our
ADC can serve as an efficient sub-converter in each stage. This
approach resolves the scalability issue by ensuring hardware cost
grows linearly, rather than exponentially, with the target resolution. A
conceptual illustration and detailed explanation of this scalable pipe-
line architecture are provided in the Supplementary Fig. 8.

The benefits of our design extend beyond component-level
comparisons to system-level improvements. Integration into CIM
systems exhibits a significant reduction in analog-to-digital conversion
overhead. Compared to energy- and area-efficient SAR ADC[49] in CIM
systems, our design achieves substantial improvements: for VGG8
implementations, as illustrated in Fig. 7, the energy contribution of
ADC components decreases from 79.8% to 22.5% of total system
energy (a 57.2% reduction), while the area overhead drops from 47.6%
to 16.9% of total chip area (a 30.7% reduction). Similar improvements
are observed in ResNet18 implementations, where ADC energy over-
head is reduced from 71.5% to 14.6% (a 56.9% reduction), and area
contribution decreases from 36.6% to 11.5% (a 25.1% reduction), as
illustrated in Supplementary Fig. 9 and Supplementary Tables 5, 6.

These comprehensive improvements across different metrics and
comparison scenarios show the advantages of our architecture,
enabling substantially higher system operating frequencies and better
energy efficiency for CIM applications. The dramatic reductions in

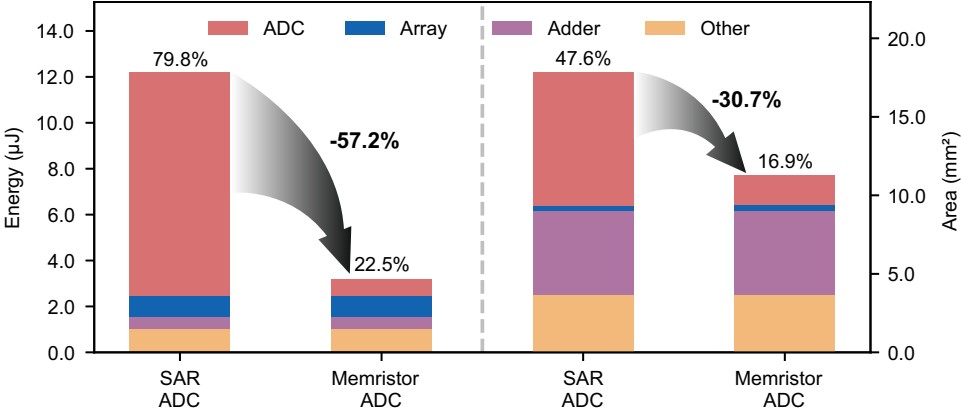

**Fig. 7 | Simulated system-level energy and area analysis of VGG8 implementation.** Energy consumption and area utilization for conventional SAR ADC and proposed memristor-based ADC in CIM systems. Stacked bars show the contributions of different components (ADC, Array, Adder, and Other) to total energy (left y-axis) and area (right y-axis). Arrows highlight the percentage reduction in energy and area for the ADC portion, demonstrating the efficiency gains of the memristor-based ADC. Detailed numerical data is provided in Supplementary Table 5.

both energy and area overhead effectively address the conversion bottleneck in CIM systems, highlighting the transformative potential of our memristor-based ADC for efficient and accurate neural network inference acceleration.

## Discussion

This work presents a memristor-based ADC architecture for neural network acceleration. Our ADC employs analog content-addressable memory cells with overlapped boundaries to establish optimized quantization thresholds, implements a super-resolution strategy to maintain competitive accuracy under practical memristor variations, and achieves a 15.1× improvement in energy efficiency and a 12.9× reduction in area compared to state-of-the-art designs. When integrated into CIM systems, it dramatically reduces ADC overhead, with energy and area reductions of 57.2% and 30.7% for VGG8, and 56.9% and 25.1% for ResNet18, respectively.

It is important to note that our ADC architecture is designed as a modular component within a larger CIM system. While this work focuses on optimizing the quantization stage, our ADC can operate synergistically with advanced techniques that address circuit-level non-idealities, such as signal margin limitations[50] and IR drop[51], as well as crossbar device-to-device variations[17,52,53]. A detailed discussion on the orthogonality and complementary nature of our approach with state-of-the-art compensation schemes is provided in Supplementary Note 4.

Building on these design principles, our ADC supports adaptive quantization while maintaining robust performance, making it particularly suitable for edge computing applications. Beyond immediate applications in inference acceleration, this design opens further possibilities for adaptive hardware-algorithm co-design in next-generation AI systems, bridging the gap between algorithmic requirements and hardware constraints.

## Methods
### Device characterization
The memristor devices were fabricated on a silicon wafer with a 300 nm thick $SiO_2$ layer. The device structure consists of a $Pt/TaO_x/Ta$ stack, where $TaO_x$ serves as the resistive switching layer with a thickness of ~5 nm. The devices are fabricated using equipment available in the clean room at The University of Hong Kong. All patterns are created through a standard lift-off process following photolithography. The bottom electrode (BE) and top electrode (TE) are composed of Cr/Pt metals deposited via an e-beam evaporator. The $TaO_x$ switching layer is formed in an ion-beam etcher operating in an oxygen-reactive sputter mode from a Ta target.

The electrical characterization of the memristor device is conducted at room temperature using a semiconductor parameter analyzer (Keysight B1500A) with the chip mounted on a probe station. To precisely control the device's conductance, we employed a customized program-and-verify scheme based on quasi-static DC sweeps. For program cycle-to-cycle variation measurement (Supplementary Fig. 1), this scheme incrementally increases the compliance current based on the difference between the measured and target conductance. The step size is determined by multiplying this difference (in μS) by an empirical scaling factor, which is set to 0.5 for the initial 10 iterations and reduced to 0.25 thereafter to enhance convergence. The process terminates after a maximum of 20 iterations. If the measured conductance overshoots the target tolerance (±5 μS), a reset sweep (starting from −1.2 V) is applied before restarting the programming process.

To gather programming statistics, target conductance values ranging from 100 μS to 400 μS were selected in 50 μS steps, with each target programmed 100 times after a reset. This procedure showed high reliability, with 91.6% of attempts converging to the target within 10 iterations and an average of 5.57 iterations required per successful programming event. For read cycle-to-cycle variation characterization (Fig. 3d), the device conductance was first set to the target values using the same program-and-verify method. Once a state was successfully programmed, its conductance was read 1000 consecutive times to assess its stability. All single-device measurements were performed on the same device to ensure consistency.

To evaluate device uniformity and statistical properties critical for practical ADC applications, we extended the analysis from a single device to an integrated 8 × 8 memristor crossbar array (64 devices) fabricated using the same process. The microscopy image in Fig. 3a shows the array test structure. All 64 devices were characterized using the identical program-and-verify scheme to assess statistical variations in programmed conductance states. Furthermore, the array was used to demonstrate analog conductance switching capabilities. The detailed results of the array-level characterization, including representative multi-device I-V curves and pattern mapping demonstrations, are provided in the Supplementary Figs. 2, 3.

### Adaptive ADC design
The reconfigurable ADC design was implemented using Spice and Cadence Virtuoso tools. The quantization cell with memristor devices was modeled and simulated. The performance metrics of the memristor-based ADC were evaluated based on the simulation results, including INL, DNL, RMS of maximum INL, and RMS of maximum DNL.

The INL, which represents the maximum deviation of the actual conversion characteristic from the ideal straight line, was calculated using:

$$\text{INL}_k = \frac{V(k) - V_{\text{best-fit}}(k)}{\text{LSB}} \quad (1)$$

where $V(k)$ is the actual voltage corresponding to output code $k$, $V_{\text{best-fit}}(k)$ is the voltage on the best-fit line. The best-fit line is determined through linear regression:

$$V_{\text{best-fit}}(k) = ak + b \quad (2)$$

where $a$ and $b$ are coefficients calculated using the least squares method.

The DNL, which measures the difference between the actual and ideal code step widths, was determined using:

$$\text{DNL}_k = \frac{V(k+1) - V(k)}{\text{LSB}} - 1 \quad (3)$$

To assess the overall nonlinearity errors, we calculated the RMS INL and RMS DNL:

$$\text{INL}_{\text{RMS}} = \sqrt{\frac{1}{N} \sum_{k=0}^{N-1} \text{INL}_k^2} \quad (4)$$

$$\text{DNL}_{\text{RMS}} = \sqrt{\frac{1}{N-1} \sum_{k=0}^{N-2} \text{DNL}_k^2} \quad (5)$$

where $N$ is the total number of ADC codes.

The design was optimized by precisely controlling the programming compliance current or pulse to adjust memristor conductance. To measure INL and DNL, a range of input voltages was applied, and the ADC's output was compared to the ideal linear response. The impact of device variation was analyzed by introducing controlled conductance variations in the memristors and observing the resulting changes in INL and DNL.

## Neural network evaluation

We evaluated our ADC strategy using two neural network architectures: VGG8 for CIFAR-10 classification and ResNet18 for ImageNet classification. The networks were implemented using a modified NeuroSim framework[49], which we enhanced with detailed ADC behavioral models to accurately capture the impact of quantization and device variations. Our simulation framework incorporates precise modeling of ADC characteristics, providing realistic performance estimates for practical CIM implementations. The networks were implemented with different ADC configurations and quantization strategies.

## Super resolution strategy

To address device variations in practical implementations, we developed the super-resolution strategy, which utilizes two ADCs with nominally identical quantization boundaries. Under memristor variations, these boundaries exhibit slight misalignments. When an input voltage falls near a quantization boundary, the two ADCs may produce different quantization results. We leverage this disagreement to identify and precisely quantize these boundary-proximate values. Specifically, when the two ADCs produce different outputs for the same input, we assign the input to the corresponding boundary value, effectively achieving super-resolution at quantization boundaries. This approach not only improves quantization accuracy but also takes advantage of inherent device variations to enhance precision.

## Implementation details for 5-bit ADC design

Our design consists of the ADC core and a decoder circuit, with their respective contributions to energy and area detailed in Fig. 6b. The estimations are based on our previous analog CAM in 16 nm technology[36]. To calculate the energy consumption, we began with the baseline analog CAM design, which consumes 0.47 fJ per operation (excluding its DAC portion). To enhance the linearity of our ADC, we expanded the memristor conductance range to 100–400 µS. This primarily affected the source driver within each Q-cell, increasing its energy consumption from 0.29 fJ to 0.58 fJ. Consequently, the total energy for the ADC core is estimated at 12.58 fJ per operation.

For the area estimation, a single Q-cell occupies an area of ~0.72 µm × 0.72 µm, or 0.52 µm², as shown in the 16-nm physical layout[36]. Therefore, the 5-bit ADC core, comprising 16 Q-cells, has an area of 8.29 µm². The decoder circuit, synthesized using a 28 nm technology library, adds an area of 16.00 µm², resulting in a total estimated ADC area of 24.29 µm². This area estimation for ADC-specific components excludes memristor write drivers since they are treated as shared system-level resources. To further validate this estimation methodology, we designed a full custom layout of a comparable 5-bit ADC in a UMC 28 nm process, which supports our scaled estimations (see Supplementary Fig. 7 for details).

The total conversion latency is estimated as 0.165 ns, which combines the core's 45 ps latency from the CAM reference and the decoder's 0.12 ns latency from synthesis. This total latency serves as the basis for our system-level timing analysis. While our estimation uses a mixed-technology approach, it represents a conservative projection. A full 16 nm implementation would likely yield further improvements in energy and area. By extrapolating these validated figures, we calculated the total energy and area metrics for various ADC configurations.

## CIM system evaluation

To comprehensively evaluate the system-level benefits of our proposed memristor-based ADC design, we conducted a detailed performance analysis using a modified NeuroSim framework[49]. The CIM architecture was implemented with memristor crossbar arrays, each configured as a 128 × 128 matrix. To optimize the balance between performance and hardware overhead, we adopted a resource-sharing strategy where every four columns share one ADC, resulting in 32 ADCs per array. All ADCs were configured with 5-bit precision (32 quantization levels) to maintain consistent comparison across different network architectures.

The complete system architecture incorporates several key components, including the core crossbar array, ADC, adder, and other peripheral circuits. The adder units include adder trees and shift register pairs for accumulating partial products. Other circuits comprise essential peripheral components such as word line decoders and their corresponding drivers, multiplexer circuits with associated decoders, and word line switch matrices. The entire system, including the conventional SAR ADC model, was evaluated under a 14 nm technology node to ensure fair comparison with state-of-the-art implementations. The detailed energy consumption and area utilization of each component were analyzed for both VGG8 and ResNet18 implementations, as presented in Fig. 7 and Supplementary Fig. 9, respectively. The comprehensive breakdown of system-level metrics is provided in Supplementary Tables 5, 6, demonstrating the significant reduction in ADC overhead achieved by our design across different network architectures.

## Robustness to voltage variations and in-situ recalibration

To assess the robustness of our ADC in practical chip environments, we conducted comprehensive simulation-based analysis of its resilience to reference voltage variations. Our evaluation systematically investigated the impact of both high-frequency dynamic power supply noise

and quasi-static DC voltage drift on ADC linearity (INL/DNL). The results show that the ADC is highly resilient to dynamic noise, as demonstrated by the 5-bit ADC which maintains INL well below 0.5 LSB even under 20 mV peak-to-peak noise.

While the analysis indicates higher sensitivity to static DC drift, this highlights a core strength of our architecture. Unlike conventional designs that require dedicated compensation circuits, our ADC can directly counteract such drifts through in-situ recalibration. Since memristor conductances define the quantization boundaries, any quasi-static voltage drift can be completely nullified by reprogramming the memristors to new conductance values corresponding to the desired thresholds under the drifted condition. This ability to directly reconfigure quantization boundaries eliminates the need for additional hardware, thus demonstrating significant advantages in efficiency and design simplicity (detailed methodology and results in Supplementary Note 3 and Supplementary Fig. 5).

## Data availability

The data that support the findings of this study are available in the public repository at https://github.com/MIKEHHQ/ReADC [54] or from the corresponding authors upon request. Source data are provided with this paper.

## Code availability

The codes used for the simulations in this study are available in the public repository at https://github.com/MIKEHHQ/ReADC [54] or from the corresponding authors upon request.

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

## Acknowledgements

This work was supported in part by the Theme-based Research Scheme (TRS) project T45-701/22-R (N.W., C.L. & W.Z.), the National Natural Science Foundation of China (62404187 (Z.L.), 62122005 (C.L.)), Croucher Foundation (C.L.), and the General Research Fund (GRF) Project (17200925 (Z.L.), 17203224 (N.W.), 17207925 (C.L.)) of the Research Grants Council (RGC), Hong Kong SAR.

## Author contributions

Z.L., C.L., and N.W. supervised the project. H.H., R.M., C.L., and N.W. contributed to the conception of the idea. H.H. performed the experiments and analyzed data under the supervision of Z.L., C.L., and N.W. Z.D. fabricated the memristors. Z.D., M.J., and R.M. provided guidance on experimental procedures. Y.R., F.L., W.M., M.P., and W.Z. contributed valuable insights through technical discussions. H.H., Z.L., C.L., and N.W. wrote the manuscript with input from all authors.

## Competing interests

The authors declare no competing interests.
