## [Transparent Peer Review file · Nature Communications]

Memristor-based adaptive analog-to-digital conversion for efficient and accurate compute-in-memory

Corresponding Author: Dr Zhengwu Liu

Version 0:

Reviewer comments:

Reviewer #1

(Remarks to the Author)

The problem that this manuscript is trying to solve is very important. Figure 1b-c are particularly clear in this respect. Having reduced the energy consumed by DAC and ADC before and after the VMM operation through the memristors is very important, I would say critical, to enable this technology and move forward.

The manuscript is well written, the figures are very clear and it presents enough data to, mainly, support its claims. For that reason, I strongly recommend its publication in Nature Communications. However, there are several points that I think the authors should clarify to avoid hype and erroneous interpretations from the readers and community in general:

1 – The proposed structure for memristor based ADC design is clear in Figure 2, but I feel it is necessary to indicate explicitly the number of components that this (Figure 2a) includes and estimate the amount of area that this would occupy in a microchips. In that sense, I find that Figure S2 is important enough to be included in the main text.

2 – The authors should very clearly indicate what has been done and what has been simulated. Because the authors are talking about circuits all the time but the only experimental images that I see are from a single isolated memristor. Did the authors fabricated one array? Can they show images of it? Please elaborate more about this. Please also add in all figure captions “Simulation of...” when it applies.

3 – Figure 3 shows very clear data but it is necessary to elaborate a bit more. Information about the variations from one device to another are necessary. How many devices were tested? In how many the resistance could be adjusted to the desired values?

4 – No endurance mention is made in the paper, and it would be necessary to understand how many times the conductance can be adjusted at the levels indicated.

5 – The data and the code statement reads “The data that support the findings of this study are available from the corresponding author upon reasonable request.” However, I think this should be put in a repository openly, as most authors publishing in Nature journals in this field do.

Reviewer #2

(Remarks to the Author)

This article presents an ADC design based on memristor. The using of ADC is one of the key issues in efficiency of memristor chip. Designing an ADC directly using memristor in a memristor-based computing-in-memory system is interesting. However, there are still several issues concerning the research topic and experimental evaluation, some of which may noticeably affect the overall strength of this work.

1. One potential concern might be the scope of the work. The focus on ADC design and quantization techniques may be more appropriate for specialized journals or

conferences in circuit design or device engineering, rather than an interdisciplinary journal such as Nature Communications. The authors are encouraged to better articulate the broader impact and interdisciplinary relevance of their work in future revisions to engage a wider audience.

2. Can you provide more details on the memristor characterization? For example, Fig. 3a lacks sufficient detail—Only the scale of device can be inferred from the microscopy image. A more complete depiction of the device's structural and material properties is recommended.

3. Figure 3d indicates a very minimal read cycle-to-cycle variation, which seems significantly lower than several state-of-the-art memristor hardware demonstrations. A more detailed description on the memristor design and the measurement would be necessary to convince the readers.

4. How is the device-to-device variation and how would it affect the performance of the ADC and the neural network tasks?

5. The benefit of the proposed adaptive quantization approach appears to diminish as the quantization bit increases. For instance, the improvement in accuracy becomes marginal (only 0.3%) at 5-bit for VGG8, and less than 4% at 6-bit for ResNet18. The authors should clarify whether this method is primarily intended for ultra-lowbit ADC applications and discuss any limitations in scaling the technique to higherbit regimes.

6. Another concern about the scalability would be, the number of cells required for ADC increases exponentially with the ADC precision. How would it affect its scalability on precision?

7. Regarding efficiency evaluation and Fig. 6, ADC performance is typically evaluated with more comprehensive metrics beyond area and energy. A detailed comparison table including resolution, sampling frequency, INL, DNL, technology node, etc., would provide a clearer and fairer benchmarking. It is also recommended that the authors extrapolate their design to higher resolutions (8-bit for example) to strengthen the comparison with other state-of-the-art ADCs.

8. There are also inconsistencies related to the reported area of the proposed memristor-based ADC. For example, the reported area of $\sim 18 \mu\text{m}^2$ for the 5-bit ADC seems questionable, given that the area of a single memristor device shown in Fig. 3 is on the order of $100 \mu\text{m}^2$. The authors should clarify how the reported area is estimated. In addition, does this estimation includes or excludes peripheral circuitry and whether the reported ADC utilizes super-resolution? A breakdown of the area estimation would be helpful for transparency.

9. Finally, there are a few minor issues, such as typos or repeated abbreviations in Line 155, Figure S1, and elsewhere. The authors are suggested to carefully proofread the manuscript and supplementary materials for consistency and correctness.

Reviewer #3

(Remarks to the Author)

Reviewer #4

(Remarks to the Author)

The paper presents a memristor-based analog-to-digital converter (ADC) architecture designed for in-memory computing (IMC) tiles, with the goal of achieving improved area and energy efficiency. The proposed design utilizes analog content-addressable memory (CAM) cells with programmable overlapped boundaries to establish optimized quantization thresholds. This architectural choice enables favorable integral non-linearity (INL) and differential non-linearity (DNL) performance for mid- and low-precision ADC implementations.

The manuscript includes system-level validation using standard benchmarks such as CIFAR-10 and ImageNet, comparing the proposed solution against existing state-of-the-art designs. The results indicate improvements in both area and energy metrics, which are promising for large-scale deployment of IMC systems.

However, I respectfully find that the overall contribution and technical depth of the work, as currently presented, are limited. While the proposed approach shows incremental advancement, it lacks sufficient novelty and rigorous analysis that would merit publication in a high-impact journal such as Nature Communications. The experimental results, though relevant,

require further elaboration and comparative evaluation to convincingly demonstrate superiority over existing methods. Below you can find the detailed comments.

On statements and claims in the manuscript:

- "Moreover, existing quantization methods struggle to handle diverse output distributions in CIM architectures". - This may not be correct as many works have dedicated techniques to mitigate these distribution effects. In [Hsu et al. JSSC'23, Khaddam et al. JSSC'22, S. K. Roy et al. JSSC'24], they have introduced several non-ideality mitigation techniques to handle output distributions.
- "ADCs occupy up to 87.8% of total energy and 75.2% of chip area" - This may not be a general statement and many recent works have presented IMC topologies with much less ADC dominance. A review paper [Hung et al. OJ-SSCS'21] includes such component-wise area and power contributions and those are quite different from what is claimed in this statement.
- "Current solutions present a clear trade-off: at low ADC precision (≤ 6 bits), uniform quantization offers hardware-friendly implementation but suffers from significant precision loss" - Many recent works have shown to have promising accuracy results using 5-bit ADCs (for instance, in [Hsu et al. JSSC'23], they use time-interleaved voltage and current comparator based ADCs with changing BL settling voltage conditions to account for changing output distributions.)

On the proposed memristor-based ADC solution:

- Is the proposed design robust to voltage variations, especially the impact of voltage fluctuations on VL and VH seems crucial? And, the impact of these voltage variations across large chip dimensions could be quite critical to analyse. The suggestion is to investigate into these impacts and report them.
- Typically, non-idealities dominated by device-to-device variations in the memory array and input-dependent IR drop on the access lines severely impact the accuracy of the IMC tiles. This work does not necessarily address those idealities. There are some recent works that tackle these issues on a chip level [Khaddam et al. JSSC'22, S. K. Roy et al. JSSC'24]. The suggestion is to investigate into these impacts and report them.
- Are the ADC related circuits such as memristor write drivers and control logic included in the ADC area calculations? The suggestion is to include the breakdown of the $18\mu\text{msq}$ ADC area number (a layout photo perhaps) and report it.
- The dimensions of the memristor device seems quite large, in the order of $10\mu\text{m}$ range (Fig. 3A). However, the area of the ADC using several such devices seems quite small (only $18\mu\text{msq}$) as claimed in the paper. A more detailed area calculations may be helpful for the readers.

On the results:

- Fig. 6A compares with non-IMC ADCs (for instance, high-speed IO ADC in [48] is signed 8-bit and is not designed for IMC tiles). Plus, [46] ADC review does not include IMC related ADC implementations, rather they target high-speed IO, DSP applications etc., and they are typically mid- to high A/D resolution implementations.]. The suggestion is to include IMC related ADC designs for a relevant comparison.
- If we compare with IMC ADC, An 8-bit ADC implementation has < 1 LSB INL and DNL with area around $400\mu\text{msq}$ [Khaddam et al. JSSC'22]. But this also includes all kinds of calibration and non-idealities compensation related circuits. These kind of works must be included in the manuscript for comparison.
- For the results shown in Fig. 4, the trend with increasing resolution seems problematic. 1-1.5% device variations already seems quite optimistic, especially if the number of such ADCs are large and still the INLs/DNLs are around 2.

References (these are, in my opinion, only a selected few to support my comments.):

- [Hsu et al. JSSC'23] Hsu, Hung-Hsi, et al. "A nonvolatile AI-edge processor with SLC–MLC hybrid ReRAM compute-in-memory macro using current–voltage-hybrid readout scheme." *IEEE Journal of Solid-State Circuits* 59.1 (2023): 116-127.
- [Khaddam et al. JSSC'22] Khaddam-Aljameh, Riduan, et al. "HERMES-core—A 1.59-TOPS/mm² PCM on 14-nm CMOS in-memory compute core using 300-ps/LSB linearized CCO-based ADCs." *IEEE Journal of Solid-State Circuits* 57.4 (2022): 1027-1038.
- [S. K. Roy et al. JSSC'24] S. K. Roy et al., "Compute SNDR-Boosted 22-nm MRAM-Based In-Memory Computing Macro Using Statistical Error Compensation," in *IEEE Journal of Solid-State Circuits*
- [Hung et al. OJ-SSCS'21] Hung, Je-Min, et al. "Challenges and trends of nonvolatile in-memory-computation circuits for AI edge devices." *IEEE Open Journal of the Solid-State Circuits Society* 1 (2021): 171-183.

Abhairaj Singh
IBM Research Zurich

Version 1:

Reviewer comments:

Reviewer #1

(Remarks to the Author)

The authors have made a good revision of the manuscript. The new part of the endurance is useful. I think the paper should be accepted.

Still, the manuscript could be further improved indicating:

1 – In how many devices has endurance been tested and what is the endurance achieved in each of them.

2 – If endurance has been tested in more than one device, may be they could include other plots in Figure S4.

3 – It would be even better if, for at least one device, the authors show the Voltage/Current versus time plot (measured with high temporal resolution) to observe how the devices switch and make sure that the resistance is stable. Showing read, write, read, erase, read for 5-10 cycles should be sufficient.

I think the authors may provide this Information if they can. But I don't need to review what they answer again, from my side this is optional (although highly recommended).

Congratulations to the authors for their excellent results

Reviewer #2

(Remarks to the Author)

The manuscript has been significantly improved and most of my previous concerns have been resolved. I don't have further questions.

Reviewer #3

(Remarks to the Author)

Reviewer #4

(Remarks to the Author)

I sincerely thank the authors for their time and effort in preparing the revised manuscript and the response letter in response to my comments.

The authors have addressed all my comments.

Response Letter to Reviewers' Comments

We sincerely appreciate the valuable time the reviewers have spent reviewing our manuscript and providing insightful comments and suggestions to help further improve the quality of our work. We believe we have addressed all of the reviewers' comments and now the paper is more rigorous in content and clearer in presentation. Our point-by-point responses to the reviewers' comments are as follows.

Reviewer #1

The problem that this manuscript is trying to solve is very important. Figure 1b-c are particularly clear in this respect. Having reduced the energy consumed by DAC and ADC before and after the VMM operation through the memristors is very important, I would say critical, to enable this technology and move forward.

The manuscript is well written, the figures are very clear and it presents enough data to, mainly, support its claims. For that reason, I strongly recommend its publication in Nature Communications. However, there are several points that I think the authors should clarify to avoid hype and erroneous interpretations from the readers and community in general:

Response:

Thank you for the very positive assessment about our work and the strong recommendation for publication. We appreciate your recognition that reducing energy consumption in DAC and ADC operations through memristor-based VMM is critical for advancing this technology. We have carefully considered all the points you raised and provide detailed clarifications below to ensure the accuracy and clarity of our claims while avoid potential hype or misinterpretation.

Comment #1:

1 – The proposed structure for memristor based ADC design is clear in Figure 2, but I feel it is necessary to indicate explicitly the number of components that this (Figure 2a) includes and estimate the amount of area that this would occupy in a microchips. In that sense, I find that Figure S2 is important enough to be included in the main text.

Response:

Thank you for this constructive suggestion. We have addressed your concerns as follows:

Fig. 2a shows a 5-bit ADC comprising 16 Q-cells arranged in a 4×4 array. Each Q-cell contains 2 memristors, 5 transistors, and 2 inverters, totaling 32 memristors, 80 transistors, and 32 inverters for the complete ADC core. Based on reference 16 nm analog CAM cell designs (**Fig. R1.1**), each Q-cell occupies approximately 0.52 μm², resulting in a core array area of 8.29 μm². Including the decoder circuitry (16.00 μm²), the total area for the 5-bit ADC is estimated at 24.29 μm².

To validate this estimation, we designed a full custom layout of the 5-bit ADC in UMC 28 nm process (**Fig. R1.2**), which shows a total area of 43.05 μm². Scaling this to 16 nm technology (scaling factor of $(16/28)^2 = 0.33$) confirms our area estimates are within the expected range.

Per your suggestion, we have moved **Fig. S2** to the main manuscript as **Fig. 7**, which demonstrates significant area advantages over conventional SAR ADCs (30.7% reduction for VGG8 and 25.1% for ResNet18 implementations). We have updated the manuscript to present these component counts and area specifications upfront in the revised version.

Fig. R1.1. Layout of analog CAM cell. The layout design of one cell in an analog CAM array using 16 nm design rules.

Fig. R1.2. Layout of proposed 5-bit memristor-based ADC in a UMC 28 nm process. This layout, created to validate our area estimation methodology, consists of 16 Q-cells arranged in a 4×4 array and an associated program driver. The total area of the design is $43.05 \mu\text{m}^2$, with the ADC core occupying $34.39 \mu\text{m}^2$ and the program driver occupying $8.67 \mu\text{m}^2$.

Revision:

1. We have included the newly designed full custom layout (**Fig. R1.2**) as **Fig. S7**.
2. In the “Implementation details for 5-bit ADC design” subsection of the **Methods**, we have added detailed area calculations showing that a single Q-cell occupies $0.52 \mu\text{m}^2$, the 16-cell array occupies $8.29 \mu\text{m}^2$, and the total ADC area including decoder is $24.29 \mu\text{m}^2$. We also clarified that write drivers are considered shared system resources:

“Our design consists of the ADC core and a decoder circuit, with their respective contributions to energy and area detailed in **Fig. 6B**. The estimations are based on our previous analog CAM in 16 nm technology. To calculate the energy consumption, we began with the baseline analog CAM design, which consumes 0.47 fJ per operation (excluding its DAC portion). To enhance the linearity of our ADC, we expanded the memristor conductance range to 100-400 μS . This primarily affected the source driver within each Q-cell, increasing its energy consumption from 0.29 fJ to 0.58 fJ. Consequently, the total energy for the ADC core is estimated at 12.58 fJ per operation.

For the area estimation, a single Q-cell occupies an area of approximately $0.72 \mu\text{m} \times 0.72 \mu\text{m}$, or $0.52 \mu\text{m}^2$, as shown in the 16-nm physical layout. Therefore, the 5-bit ADC core, comprising 16 Q-cells, has an area of $8.29 \mu\text{m}^2$. The decoder circuit, synthesized using a 28 nm technology library, adds an area of $16.00 \mu\text{m}^2$, resulting in a total estimated ADC area of $24.29 \mu\text{m}^2$. This area

estimation for ADC-specific components excludes memristor write drivers since they are treated as shared system-level resources. To further validate this estimation methodology, we designed a full custom layout of a comparable 5-bit ADC in a UMC 28 nm process, which supports our scaled estimations (see **Fig. S7** for details).

The total conversion latency is estimated as 0.165 ns, which combines the core's 45 ps latency from the CAM reference and the decoder's 0.12 ns latency from synthesis. This total latency serves as the basis for our system-level timing analysis. While our estimation uses a mixed-technology approach, it represents a conservative projection. A full 16 nm implementation would likely yield further improvements in energy and area. By extrapolating these validated figures, we calculated the total energy and area metrics for various ADC configurations.”

3. We have substantially revised the hardware efficiency evaluation and all related figures to reflect the updated area values. The specific changes are:
 - a. In the **Abstract** and the **Introduction**, the overall cost improvement is updated to “Compared to state-of-the-art ADCs, our design achieves a 15.1× improvement in energy efficiency and a 12.9× reduction in area.”. The system-level area overhead reduction is updated to “the integration of our ADC into CIM systems reduces the energy and area overhead by 57.2% and 30.7% respectively for the CIFAR-10 (VGG8) task, and by 56.9% and 25.1% for the ImageNet (ResNet18) task.”
 - b. In the “Hardware efficiency evaluation of memristor-based ADC” section, the text now reflects the updated system-level area reductions for both VGG8 (from 47.6% to 16.9%, a 30.7% reduction) and ResNet18 (from 36.6% to 11.5%, a 25.1% reduction).
 - c. We have updated **Fig. 6** with the corrected area estimations. The scatter plot in **Fig. 6A** now reflects the new area of 24.29 μm^2 , and the area breakdown in **Fig. 6B** is updated to show the ADC core (8.29 μm^2) and the decoder (16.00 μm^2).
 - d. We have moved the system-level analysis for VGG8 from the **Supplementary Information** to the main text as the new **Fig. 7** (formerly **Fig. S2**) and updated it with the updated area data.
 - e. We have updated **Fig. S9** (formerly **Fig. S3**) for the ResNet18 system-level analysis with the updated area data.
 - f. We have updated **Table S5** and **Table S6** in the **Supplementary Information** with the updated area breakdown for the VGG8 and ResNet18 implementations, respectively.
-

Comment #2:

2 – The authors should very clearly indicate what has been done and what has been simulated. Because the authors are talking about circuits all the time but the only experimental images that I see are from a single isolated memristor. Did the authors fabricated one array? Can they show images of it? Please elaborate more about this. Please also add in all figure captions “Simulation of...” when it applies.

Response:

Thank you for this critical observation. You are absolutely correct that clear distinction between experimental and simulated results is essential for rigorous evaluation of our work. We have addressed all three aspects of your concern comprehensively.

Regarding the distinction between experimental and simulated results, all figure captions now explicitly indicate the nature of each result - simulated results (**Figs. 2, 4, 5, 6**) begin with “Simulated...” and “Estimated...” while experimental data (**Fig. 3**) is labeled as “Experimental characterization...”. For the fabricated array with supporting images, we have conducted extensive characterization on a fabricated 8×8 memristor array (64 devices). This includes optical micrographs of the fabricated array structure (**Fig. R1.3**), statistical analysis of device-to-device (D2D) variations showing a standard deviation of 2.73 μS for programmed states, which validates the variation parameters used in our simulations (**Fig. R1.4**), representative I-V switching curves from multiple devices demonstrating consistent behavior (**Fig. R1.5**), and complex pattern programming capability confirming functional yield and individual addressability of memristors (**Fig. R1.6**).

Crucially, while the ADC circuit design, neural network performance, and hardware efficiency analyses are evaluated through simulations, these simulations are rigorously parameterized and validated by our extensive experimental data from both single devices and the newly characterized array.

Fig. R1.3. Optical micrographs of the fabricated 8×8 memristor array (64 devices). The devices utilize a Pt/TaO_x/Ta material stack on Si/SiO₂ substrate. Images show (left) the complete array with contact pads, (middle) magnified view of the crossbar structure, and (right) individual device cells. TE: top electrode; BE: bottom electrode.

Fig. R1.4. Experimental device-to-device (D2D) variation measured across all 64 devices of the 8×8 array. (A) The standard deviation (σ) of the programmed conductance is shown for seven distinct target states. The variation remains consistently stable, with σ values around 2.73 μS . **(B)** Conductance distributions for the seven target states (100, 150, 200, 250, 300, 350, and 400 μS), achieved using a program-and-verify scheme. The distinct, well-separated peaks confirm that multiple states can be reliably programmed across the array.

Fig. R1.5. I-V switching characteristics of memristor devices from the 8×8 array. Ten consecutive voltage sweep cycles are shown for five devices at different array locations, demonstrating consistent bipolar switching behavior and uniform performance across the fabricated array.

Fig. R1.6. Demonstration of multi-state conductance programming on the 8×8 memristor array. Seven different letter patterns (‘H’, ‘K’, ‘U’, ‘Re’, ‘A’, ‘D’, ‘C’) were successfully programmed across the array with the color map representing measured conductance values. The clear formation of all patterns confirms high device yield and precise conductance state control across individual devices.

Revisions:

1. We have added the micrographs of our fabricated 8×8 memristor array (**Fig. R1.3**), the D2D variation statistics (**Fig. R1.4**), the representative I-V curves from multiple devices (**Fig. R1.5**), and the pattern programming results (**Fig. R1.6**) to main manuscript as new **Figs. 3A,C** and **Figs. S2-S3**, respectively.
2. For completeness, the original figures showing single-device variation characterization have been moved to the **Supplementary Information**. The original single-device micrograph (previously **Fig. 3A**) is removed. The original C2C variation plot (previously **Fig. 3C**) is now **Fig. S1**.
3. We revised the text in the “Experimental benchmarking of uniform quantization ADC” section to describe these new experimental results from the 8×8 array. On page 7-8, we added:

“To experimentally demonstrate our ADC design, we fabricated and characterized an 8×8 memristor array comprising 64 devices (**Fig. 3A**). The devices exhibit stable, multi-level conductance states controlled by programming compliance current (**Fig. 3B**, which is fundamental for establishing quantization boundaries. Statistical analysis of the array revealed device-to-device variation with a standard deviation of 2.73 μS for programmed states in the range of 100-400 μS , corresponding to <1% relative variation (**Fig. 3C**). Excellent read stability was demonstrated with only 0.14 μS (0.047%) conductance variation over 1000 consecutive cycles (**Fig. 3D**). These experimental characterizations (including **Fig. S1-S3**) validate the variation parameters to be used in our following simulation evaluations. Furthermore, endurance testing confirmed robust switching over 3×10^7 cycles within the required conductance window (**Fig. S4**). For ADC applications where write operations are infrequent, this endurance projects to a device lifetime of

approximately 87 years under a conservative estimate of 1,000 daily updates, ensuring excellent long-term reliability.”

4. We have revised the captions for all relevant figures in the main manuscript to explicitly state whether the results are from experiments or simulations:
 - a. **Fig. 1** caption is revised to “(A) Output distributions from simulated crossbar arrays...”
 - b. **Fig. 2** caption is revised to “(B) Simulated responses to the input voltage...”
 - c. **Fig. 3** caption is revised to “Experimental characterization of the fabricated memristor device.”
 - d. **Fig. 4** caption is revised to “Simulated performance metrics of the memristor-based ADC.”
 - e. **Fig. 5** caption is revised to “Simulation analysis of adaptive quantization and network performance with our ADC.”
 - f. **Fig. 6** caption is revised to “Estimated energy and area comparison of memristor-based ADC.”
 - g. **Fig. 7** and **S9** (formerly **Figs. S2-S3**): their captions are revised to “Simulated system-level energy and area analysis ...”
 - h. **Table S2** caption is revised to “Simulated memristor ADC linearity metrics”
 - i. **Table S4** caption is revised to “Detailed performance metrics comparing the proposed simulated ADC...”
 - j. **Table S5-S6** caption is revised to “... network on a simulated CIM system...”
-

Comment #3:

3 – Figure 3 shows very clear data but it is necessary to elaborate a bit more. Information about the variations from one device to another are necessary. How many devices were tested? In how many the resistance could be adjusted to the desired values?

Response:

We thank the reviewer for highlighting the need for more extensive device variation data. In response, we have performed additional experiments:

First, we fabricated and tested an 8×8 memristor array (**Fig. R1.3**), and characterized all 64 devices. We have included a statistical analysis (**Fig. R1.4**) of the conductance variation across these devices, when programmed to target states, which shows a standard deviation of 2.73 μS . The distribution is consistent with the cycle-to-cycle variation data from the single device presented in the original manuscript (**Fig. 3C**), providing confidence in its representativeness. This dataset confirms that the conductance levels required for our ADC operation (e.g., 100-400 μS) are achievable across multiple devices.

Second, we now provide representative IV switching curves (each with 10 cycles) from 5 distinct devices located at different positions (Top Left, Top Right, Center, Bottom Left, and Bottom Right) within the array (**Fig. R1.5**). These curves, obtained under identical programming conditions, further illustrate the device-to-device operational consistency and variability.

Third, from our 8×8 array, we observed that all 64 devices could be reliably programmed into multiple distinct conductance states within the target window required for our ADC operation. The pattern programming experiments (“H, K, U, Re, A, D, C” as shown in **Fig. R1.6**) further demonstrate successful multi-device programmability within the array.

The data from **Fig. 3** in the original manuscript was indeed from a single, representative device, subjected to multiple programming cycles. The new array data now provides a broader understanding of D2D variations, which has been incorporated into our analysis and discussion.

Revisions:

1. We have incorporated our new experimental data from a fabricated 8×8 memristor array (64 devices tested) directly into the main manuscript and **Supplementary Information**.
2. **Fig. 3** of the main manuscript has been revised to prominently feature the array characterization: **Fig. 3A** now presents the micrographs of the 8×8 array. **Fig. 3C** shows the D2D variation statistics from all 64 devices.
3. Additional detailed characterization from the array is provided in the **Supplementary Information**:
 - 1) We added **Fig. R1.5** (I-V curves from 5 distinct devices) as **Fig. S2**.
 - 2) We added **Fig. R1.6** (pattern programming on the 8x8 array) as **Fig. S3**.
4. We have updated the text in the “Experimental benchmarking of uniform quantization ADC” section (Page 7) to clearly state that 64 devices were characterized and to discuss the results, for example: “...we fabricated and characterized an 8×8 memristor array comprising 64 devices (**Fig. 3A**)...”
5. We also revised the **Methods** section (“Device characterization”, Page 17) to include details of the array characterization process: “To evaluate device uniformity and statistical properties critical for practical ADC applications, we extended the analysis from a single device to an integrated 8×8

memristor crossbar array (64 devices) fabricated using the same process. The microscopy image in **Fig. 3A** shows the array test structure. All 64 devices were characterized using the identical program-and-verify scheme to assess statistical variations in programmed conductance states. Furthermore, the array was used to demonstrate analog conductance switching capabilities. The detailed results of the array-level characterization, including representative multi-device I-V curves and pattern mapping demonstrations, are provided in the **Figs. S2-3.**”

Comment #4:

4 – No endurance mention is made in the paper, and it would be necessary to understand how many times the conductance can be adjusted at the levels indicated.

Response:

We thank the reviewer for this important question regarding endurance, which is critical for assessing the practical viability of our device. We have now conducted extensive endurance testing on our memristor devices and provide a more detailed justification for their long-term reliability. Our devices have demonstrated robust endurance (**Fig. R1.7**), achieving over 31,755,000 SET/RESET cycles while maintaining the ability to be programmed within the target conductance window (100-400 μS).

Fig. R1.7. Experimental endurance characterization of the Pt/TaO_x/Ta memristor device. The device was subjected to repeated SET (target > 400 μS) and RESET (target < 100 μS) cycles. The high-conductance state (HCS, red) and low-conductance state (LCS, blue) are plotted against the cycle number on a logarithmic scale. The dashed lines indicate the operational conductance window required for the ADC application. The device demonstrates robust endurance, successfully completing over 3×10^7 cycles while maintaining a clear and stable distinction between the HCS and LCS states within the required operational window (Although some ultrahigh conductance states were observed during the final period, it can always recover to LCS after several reset trials), confirming its reliability for the intended application.

The testing procedure involved applying alternating SET (to above 400 μS) and RESET (to below 100 μS) pulses and verifying the state after each cycle. While some works¹⁻³ have reported closed (10^7) and even higher endurance (10^{12}) in literature, this demonstrated performance is substantial and more than sufficient for the intended ADC application.

For ADC applications, especially in inference tasks, the write operations (adjusting quantization boundaries) are far less frequent than read operations. For adaptive quantization, boundaries might be updated when a new model is loaded or when switching between layers with significantly different data distributions.

To provide a quantitative assessment of the device's lifespan under a highly demanding scenario, we performed a lifetime estimation. We assume a deliberately conservative and frequent update rate

of 1,000 times per day, which is a stress condition that far exceeds typical use cases, to project the operational lifetime. The calculation is as follows:

$$\text{Operational Lifetime} = \frac{\text{Total Endurance Cycles}}{\text{Updates per Day} \times \text{Days per year}} = \frac{31,755,00 \text{ cycles}}{1,000 \frac{\text{cycles}}{\text{day}} \times 365.25 \frac{\text{days}}{\text{year}}} \\ \approx 87 \text{ years}$$

This calculated lifetime of approximately 87 years demonstrates that the device endurance is more than sufficient for the intended application, providing a very long operational lifetime even under strenuous, accelerated conditions. This robust performance ensures the long-term reliability of our ADC in practical CIM systems.

Revisions:

1. We have added **Fig. R1.7** as **Fig. S4** in the revised **Supplementary Information**.
 2. We have added a new paragraph to the the “Experimental benchmarking of uniform quantization ADC” section: “Furthermore, endurance testing confirmed robust switching over 3×10^7 cycles within the required conductance window (**Fig. S4**). For ADC applications where write operations are infrequent, this endurance projects to a device lifetime of approximately 87 years under a conservative estimate of 1,000 daily updates, ensuring excellent long-term reliability.”
-

Comment #5:

5 – The data and the code statement reads “The data that support the findings of this study are available from the corresponding author upon reasonable request.” However, I think this should be put in a repository openly, as most authors publishing in Nature journals in this field do.

Response:

We fully agree with the reviewer on the importance of open data and code sharing. We have revised our “Data Availability” and “Code Availability” statements to reflect our commitment to making all relevant data and code publicly available in a repository upon acceptance of the manuscript, in accordance with Nature Portfolio policies. The updated statement now reads: “The data that support the findings of this study, including raw experimental data and simulation code, will be deposited in a publicly accessible repository (<https://github.com/MIKEHHQ/ReADC>). Source data are provided with this paper.”

Revisions:

1. We updated our “Data Availability” and “Code Availability” statements accordingly.

Reviewer #2

This article presents an ADC design based on memristor. The using of ADC is one of the key issues in efficiency of memristor chip. Designing an ADC directly using memristor in a memristor-based computing-in-memory system is interesting. However, there are still several issues concerning the research topic and experimental evaluation, some of which may noticeably affect the overall strength of this work.

Response:

We thank you for recognizing the importance of our research topic and for the constructive feedback. We have addressed each concern with additional details, experiments, and clarifications.

Comment #1:

1. One potential concern might be the scope of the work. The focus on ADC design and quantization techniques may be more appropriate for specialized journals or conferences in circuit design or device engineering, rather than an interdisciplinary journal such as Nature Communications. The authors are encouraged to better articulate the broader impact and interdisciplinary relevance of their work in future revisions to engage a wider audience.

Response:

We thank the reviewer for this comment regarding the scope, and we agree that we should better articulate the interdisciplinary relevance and broad impact of our work. We have revised our **Introduction** to reflect this.

We believe our work possesses significant interdisciplinary relevance that aligns well with *Nature Communications*' mission. Compute-in-memory is inherently an interdisciplinary field, bridging device physics, circuit design, computer architecture, and AI algorithms (Aguirre et al.⁴, *Nature Communications*, 2024). Our work exemplifies this convergence by targeting the ADC, a critical system-level bottleneck, with a holistic, co-design approach.

Furthermore, we position our work as a pioneering step towards future, highly-integrated CIM systems. The continuous drive to accommodate large-scale models is pushing CIM technology towards ultra-high-density, memristor-centric paradigms⁴. In these emerging architectures, a "full memristor-based" design philosophy, where memristors form not only the computational core but also critical peripheral circuits such as the ADC, becomes crucial for achieving a self-consistent and highly-integrated system, thereby mitigating the significant overhead and design complexity inherent in hybridizing disparate analog device technologies.

Our work is a direct exploration of this important concept. By designing an ADC directly using memristors, we are not only addressing a present-day bottleneck in an area- and energy-efficient manner but also proposing a new design philosophy for next-generation AI hardware. This has broad implications for the future of CIM, moving beyond hybrid CMOS-memristor systems towards more deeply integrated solutions.

Therefore, our contribution extends beyond a specialized circuit design; it provides a tangible solution to a current system-level problem while simultaneously offering a forward-looking perspective on the co-evolution of devices and architectures for AI.

Revisions:

1. We have revised the **Introduction** section to better emphasize the broader impact and system-level significance of our work. Specifically, on page 2, in the second paragraph, we have modified the text to state: “Current ADCs, as fundamental components for signal conversion, consume excessive energy and circuit area, fundamentally constraining the scalability of CIM systems as neural network complexity grows.” This change positions our work within the broader context of CIM system advancement rather than as a narrow circuit design problem.
 2. We have streamlined the transition to our solution in the third paragraph (page 3) by replacing the original opening sentence with: “To address these fundamental challenges, we introduce a memristor-based ADC that leverages the intrinsic programmability of memristive devices to enable adaptive quantization directly in hardware. It uses analog content-addressable memory (CAM) cells to implement this approach in CIM systems, significantly reducing quantization errors compared to uniform approaches.” This revision more clearly connects our approach to the fundamental challenges in CIM, emphasizing how our memristor-based design leverages intrinsic device properties to address system-level requirements.
-

Comment #2:

2. Can you provide more details on the memristor characterization? For example, Fig. 3a lacks sufficient detail—Only the scale of device can be inferred from the microscopy image. A more complete depiction of the device's structural and material properties is recommended.

Response:

We thank the reviewer for pointing out the need for greater detail on our device fabrication and structure. We have substantially revised the “Device Characterization” section in the **Methods** to provide a more comprehensive description. The updated section now includes:

1. We explicitly state the full device structure as a Pt/TaO_x/Ta stack on a Si/SiO₂ substrate, with TaO_x being the ~5 nm thick switching layer.
2. We have added a description of the fabrication workflow, clarifying that the device patterns were created using photolithography and a lift-off process. We also detail that the metal electrodes (Cr/Pt) were deposited via e-beam evaporation, and the crucial TaO_x layer was formed using oxygen-reactive ion-beam sputtering.
3. We clarify in the **Methods** that the microscopy image in **Fig. 3A** of the main manuscript shows a top-down view of the 8×8 crossbar test array. We have removed the micrograph of the single device (previous **Fig. 3A**). This is supported by extensive new array-level characterization, including a statistical analysis of device-to-device variation across all 64 devices, representative I-V curves from multiple device locations, and a successful demonstration of pattern programming, all of which have been integrated into the manuscript and **Supplementary Information** (see our response to your Comment #3 for details).

We believe these additions now provide a complete picture of the device’s structural and material properties, directly addressing the reviewer’s concern. While our specific TaO_x-based device serves as a robust proof-of-concept, we also reiterate that our ADC architecture is fundamentally adaptable to other memristor technologies that offer similar analog programmability.

Revisions:

1. We have substantially expanded the “Device characterization” subsection in **Methods** (Page 16) to include comprehensive details about the device structure, materials (Pt/TaO_x/Ta stack), and fabrication process (photolithography, lift-off, e-beam evaporation, and ion-beam sputtering). We added: “The memristor devices were fabricated on a silicon wafer with a 300 nm thick SiO₂ layer. The device structure consists of a Pt/TaO_x/Ta stack, where TaO_x serves as the resistive switching layer with a thickness of ~5 nm. The devices are fabricated using equipment available in the clean room at The University of Hong Kong. All patterns are created through a standard lift-off process following photolithography. The bottom electrode (BE) and top electrode (TE) are composed of Cr/Pt metals deposited via an e-beam evaporator. The TaO_x switching layer is formed in an ion-beam etcher operating in an oxygen-reactive sputter mode from a Ta target.

...

To evaluate device uniformity and statistical properties critical for practical ADC applications, we extended the analysis from a single device to an integrated 8×8 memristor crossbar array (64 devices) fabricated using the same process. The microscopy image in **Fig. 3A** shows the array test structure. All 64 devices were characterized using the identical program-and-verify scheme to assess statistical variations in programmed conductance states. Furthermore, the array was used to demonstrate analog conductance switching capabilities. The detailed results of the array-level

characterization, including representative multi-device I-V curves and pattern mapping demonstrations, are provided in the **Figs. S2-3.**”

2. To provide clear visual context, we have updated the figures as follows: 1)The micrographs of the new 8×8 memristor array are now presented in the main manuscript as **Fig. 3A**.The micrograph of the large-scale single-device test structure has been removed from the manuscript.
-

Comment #3:

3. Figure 3d indicates a very minimal read cycle-to-cycle variation, which seems significantly lower than several state-of-the-art memristor hardware demonstrations. A more detailed description on the memristor design and the measurement would be necessary to convince the readers.

Response:

We thank the reviewer for this insightful observation. We have elaborated on the design, programming, and measurement procedures in the “Device Characterization” section of the Methods to provide further conviction:

1. As mentioned, our Pt/TaO_x/Ta devices are optimized for stable switching. The relatively wide conductance range our devices can achieve (e.g., 100-400 μ S for ADC operation) means that even small absolute current fluctuations during read translate to a very small percentage variation of the set conductance state, especially for higher conductance values.
2. The states in **Fig. 3D** were set using a meticulous program-and-verify scheme (detailed in Methods) to achieve the target conductance precisely. Once set, the read operations are performed at a much lower voltage that does not disturb the state, contributing to high read stability.
3. Our new device-to-device variation data from the 8 \times 8 array (**Fig. R2.1-R2.2**, showing a 2.73 μ S standard deviation for programmed states across different devices) and the multi-cycle IV curves from 5 different devices (**Fig. R2.3**) demonstrate consistent switching behavior across multiple devices. This device-to-device consistency, combined with the low cycle-to-cycle variation shown in the original **Fig. 3D**, confirms the overall reliability of our memristor technology.

We believe that this combination of optimized device design, careful programming, and precise characterization techniques provides a robust foundation for the reliable operation of our memristor-based ADC.

Fig. R2.1. Optical micrographs of the fabricated 8 \times 8 memristor array (64 devices). The devices utilize a Pt/TaO_x/Ta material stack on Si/SiO₂ substrate. Images show (left) the complete array with contact pads, (middle) magnified view of the crossbar structure, and (right) individual device cells. TE: top electrode; BE: bottom electrode.

Fig. R2.2. Experimental device-to-device (D2D) variation measured across all 64 devices of the 8×8 array. (A) The standard deviation (σ) of the programmed conductance is shown for seven distinct target states. The variation remains consistently stable, with σ values around $2.73 \mu\text{S}$. (B) Conductance distributions for the seven target states (100, 150, 200, 250, 300, 350, and 400 μS), achieved using a program-and-verify scheme. The distinct, well-separated peaks confirm that multiple states can be reliably programmed across the array.

Fig. R2.3. I-V switching characteristics of memristor devices from the 8×8 array. Ten consecutive voltage sweep cycles are shown for five devices at different array locations, demonstrating consistent bipolar switching behavior and uniform performance across the fabricated array.

Revisions:

1. We have moved our new, comprehensive experimental data on D2D variation, measured across a fabricated 8×8 array, into the main manuscript as **Fig. 3C**. This prominently showcases the low variation and device consistency.
2. We added a new paragraph in the “Experimental benchmarking of uniform quantization ADC” section (Page 6-7) to explicitly connect this experimental D2D data to our simulation parameters.

The text now includes: “To experimentally demonstrate our ADC design, we fabricated and characterized an 8×8 memristor array comprising 64 devices (**Fig. 3A**). The devices exhibit stable, multi-level conductance states controlled by programming compliance current (**Fig. 3B**), which is fundamental for establishing quantization boundaries. Statistical analysis of the array revealed device-to-device variation with a standard deviation of 2.73 μS for programmed states in the range of 100-400 μS , corresponding to <1% relative variation (**Fig. 3C**). Excellent read stability was demonstrated with only 0.14 μS (0.047%) conductance variation over 1000 consecutive cycles (**Fig. 3D**). This experimental characterization validates the variation parameters to be used in our following simulation evaluations.”

3. The full set of new array characterization figures from this response letter have been added to the revised manuscript and **Supplementary Information**: 1) **Fig. R2.1** (array micrographs) is now **Fig. 3A** in the main manuscript. 2) **Fig. R2.2** (D2D variation) is now **Fig. 3C** in the main manuscript. 3) **Fig. R2.3** (multi-device I-V curves) is now **Fig. S2**.
 4. The original plots detailing single-device read stability and C2C programming variation are preserved to provide a complete picture: 1) The read C2C variation remains in the main manuscript as **Fig. 3D**. 2) The program C2C variation (original **Fig. 3C**) has been moved to **Fig. S1**.
 5. We revised **Methods** section (“Device characterization”, Page 17) to detail the measurement procedures for the 8×8 array.
-

Comment #4:

4. How is the device-to-device variation and how would it affect the performance of the ADC and the neural network tasks?

Response:

We appreciate the reviewer’s question regarding D2D variation and its impact on performance. To address this, we have now conducted comprehensive experimental characterization of D2D variation across an 8×8 array (64 devices). Our measurements show a D2D variation with a standard deviation of 2.73 μS when devices are programmed to target states within the operational range of 100 μS to 400 μS (Fig. R2.2).

Therefore, the measured D2D standard deviation represents a relative variation of less than 1% (2.73 μS / 300 μS \approx 0.9%). This comprehensive D2D variation is consistent with the cycle-to-cycle (C2C) variation of <1% observed on a single device, giving us high confidence in the stability and uniformity of our devices.

This new experimental finding strongly validates the variation parameters used in our original simulations. Our simulations in Fig. R2.4 (revised Fig. 4 in the manuscript) comprehensively analyze the impact of device variation on ADC linearity (INL/DNL). As the plots show, at a 1% variation level, a conservative upper bound for our experimentally measured D2D variation, the ADC performance is excellent. Specifically, for a 5-bit ADC, the simulated RMS of maximum INL and DNL are approximately 0.319 LSB and 0.419 LSB, respectively, well below the 1 LSB threshold required for robust operation.

Fig. 2.4. Simulated performance metrics of the memristor-based ADC. (A) INL of 4-bit ADC under 1% variation across 16 digital codes, maximum 0.215 LSB. (B) DNL under the same conditions, maximum 0.322 LSB.

(C-D) RMS analysis of maximum INL (C) and DNL (D) under various device variations for ADC resolutions from 2 to 6 bits, illustrating the ADC’s sensitivity to such variations.

Crucially, our super-resolution strategy, introduced and evaluated in the neural network performance section (Figs. 5E-F), is specifically designed to mitigate the impact of such D2D variations. It effectively leverages the inherent differences between paired ADCs (arising from D2D variations) to achieve finer effective quantization steps. This strategy not only helps recover accuracy loss due to variations but also, in some cases (e.g., 4-bit ResNet18), even surpasses the theoretical adaptive quantization baseline, demonstrating its efficacy in transforming device non-idealities into a potential advantage at the system level. We have updated the relevant sections of the manuscript to include the new experimental D2D data and further elaborate on its impact and how our super-resolution strategy addresses it in the context of neural network applications.

Revisions:

1. We have moved our new experimental data on D2D variation, measured across the fabricated 8×8 array, into the main manuscript as Fig. 3C. This prominently showcases the low variation ($\sigma = 2.73 \mu\text{S}$) and device consistency. We add: “Statistical analysis of the array revealed device-to-device variation with a standard deviation of 2.73 μS for programmed states in the range of 100-400 μS , corresponding to <1% relative variation (Fig. 3C).”
2. In the “Experimental benchmarking of uniform quantization ADC” section, we added a new paragraph explicitly linking this experimental data to our simulation parameters. The text now includes: “This experimental characterization validates the variation parameters to be used in our following simulation evaluations.”
3. The “Device characterization” subsection of **Methods** now includes a detailed description of the fabrication and characterization process for the 8×8 memristor array: “To evaluate device uniformity and statistical properties, which are critical for practical ADC applications, we extended our analysis from single devices to an integrated array. An 8×8 memristor crossbar array (64 devices) was fabricated using the same process. All 64 devices were characterized using the identical program-and-verify scheme described previously to assess statistical variations in programmed conductance states. Furthermore, the array was used to demonstrate analog conductance switching capabilities. The detailed results of the array-level characterization, including representative multi-device I-V curves and pattern mapping demonstrations, are provided in the Figs. S2-3.”

Comment #5:

5. The benefit of the proposed adaptive quantization approach appears to diminish as the quantization bit increases. For instance, the improvement in accuracy becomes marginal (only 0.3%) at 5-bit for VGG8, and less than 4% at 6-bit for ResNet18. The authors should clarify whether this method is primarily intended for ultra-low bit ADC applications and discuss any limitations in scaling the technique to higher bit regimes.

Response:

Thanks for your insightful observation. Your comment is correct that the performance gain of adaptive quantization is task-dependent, and we agree that this highlights a fundamental and important aspect of our work. It gives us a welcome opportunity to clarify the value proposition of our approach across different network complexities and bitwidths.

The core of our argument is that the advantage of adaptive quantization extends to higher bitwidths for more complex tasks. For simpler networks like VGG8 on CIFAR-10, the accuracy improvement becomes modest at 5-bit (89.9% uniform vs. 90.2% adaptive). This is because for a relatively simple task, a 5-bit uniform ADC already provides sufficient granularity to capture the output distribution reasonably well. The most significant gains are indeed at lower bitwidths (e.g., +55.1% at 3-bit and +36.6% at 4-bit), where standard uniform quantization fails catastrophically. For complex networks like ResNet18 on ImageNet, our adaptive ADC demonstrates its true strength and scalability to higher precision. The output distributions from a deeper, more complex network are far more varied and challenging to quantize. Here, the benefits of our adaptive approach are not only significant but crucial, even at 6-bit precision.

Contrary to the suggestion that our method’s benefits diminish, our results show a **+17.3%** absolute accuracy improvement at 6-bit with our super-resolution strategy (65.5% vs. 48.2% for uniform). Even without the SR strategy, the adaptive ADC alone provides a substantial **+10.8%** accuracy gain (59.0% vs. 48.2%).

This clearly demonstrates that our adaptive ADC is not limited to “ultra-low bit” applications. Instead, its effective precision range scales with the complexity of the neural network. For complex, real-world models like ResNet18, 6-bit precision falls squarely within the regime where our adaptive approach offers a transformative advantage.

Regarding the limitations in scaling to even higher bit regimes (e.g., >8 bits), we have already considered this. As detailed in our response to your **Comment #6**, for very high-precision requirements, we propose a hybrid pipeline architecture that leverages our ADC as an efficient sub-converter. This approach overcomes the exponential scaling of a single-stage design and provides a clear and viable path for future high-resolution implementations.

Revisions:

We have expanded our analysis in the “Adaptive quantization ADC for efficient neural networks” subsection in **Methods** to clarify the relationship between network complexity and the benefits of adaptive quantization. Specifically, on page 12, we added a new paragraph to explicitly address the trend of adaptive quantization performance with increasing network complexity, and to highlight the significant accuracy gains at 6-bit for the ResNet18 task: **“The above results also reveal that the performance advantage of our adaptive quantization scales with network complexity. For simpler networks like VGG8, benefits are most pronounced at very low bitwidths (+55.1% at 3-bit), while the**

accuracy gap narrows at 5-bit as uniform quantization becomes adequate. Conversely, complex networks like ResNet18 on ImageNet generate more diverse output distributions, extending the value of adaptivity to higher bitwidths. Rather than exhibiting diminishing returns, our approach delivers a substantial +17.3% absolute accuracy improvement at 6-bit (65.5% with SR vs. 48.2% for uniform), demonstrating effectiveness well beyond the ‘ultra-low bit’ regime..”

Comment #6:

6. Another concern about the scalability would be, the number of cells required for ADC increases exponentially with the ADC precision. How would it affect its scalability on precision?

Response:

We acknowledge the reviewer’s concern regarding the exponential scaling of Q-cells with ADC precision. As discussed in our response to **Comment #5**, this exponential relationship (2^{n-1} Q-cells for an n-bit ADC) is indeed a fundamental aspect of our design. This makes it highly efficient for its target application in the mid-to-low bitwidth range (3-6 bits) but presents a scalability challenge for very high-resolution (e.g., >8 bits) single-stage implementations.

To address this scaling challenge, we propose integrating our proposed ADC (termed ReADC, reflecting its use of resistive memory elements and its reconfigurability) into a hybrid pipeline architecture (**Fig. R2.5**). This approach leverages the established multi-stage pipeline ADC methodology, where each stage resolves a small number of bits and passes a residue to the next.

The operation follows a straightforward sequence as illustrated in **Fig. R2.5A**. The analog input voltage (V_{in}) is first sampled and held, then quantized by an n-bit ReADC acting as the sub-ADC. The n-bit digital output feeds into an n-bit DAC to create an analog representation, which is subtracted from the original input to generate the quantization residue. This small residue voltage is amplified by a factor of 2^n and passed to the subsequent stage. **Fig. R2.5B** shows how cascading four 4-bit stages achieves 16-bit total resolution when n-bit equals to 4-bit.

This pipeline approach transforms our exponential scaling challenge into a linear one. Hardware cost now scales linearly with the number of stages rather than exponentially with total bits. Additionally, each sub-ADC only needs linearity over its small n-bit range, making the system more robust to memristor variations while potentially reducing power consumption through narrower conductance programming ranges.

Fig. R2.5. Proposed pipeline ADC architecture for high-resolution applications using proposed ADC (ReADC) as the sub-ADC. (A) Single pipeline stage architecture. The stage consists of an n-bit ReADC sub-converter, a DAC, subtraction circuit (Σ), and gain amplifier ($\times G$). It processes input V_{in} to produce an n-bit digital output and passes amplified residue V_r for the next stage. (B) Four-stage pipeline example achieving $(4 \times n)$ -bit total resolution by cascading four n-bit stages. The Digital Correction & Combiner block integrates outputs from all stages into the final high-resolution word.

Revision:

1. We have added **Fig. R2.5** as **Fig. S8** in revised version to illustrate a scalable pipeline architecture.
 2. To address the scalability of our design for higher resolutions, we have introduced a discussion on a hybrid pipeline architecture within the “Hardware efficiency evaluation of memristor-based ADC” subsection in **Methods**. On Page 14, we added a new paragraph to introduce the scalable pipeline architecture: “While our single-stage adaptive memristor-based ADC excels in latency and energy for low-to-mid bitwidths, its flash-like architecture faces a scalability challenge for achieving very high resolutions due to the exponential growth of Q-cells. For such applications, a promising direction is to employ a multi-stage pipeline architecture, where our ADC can serve as an efficient sub-converter in each stage. This approach resolves the scalability issue by ensuring hardware cost grows linearly, rather than exponentially, with the target resolution. A conceptual illustration and detailed explanation of this scalable pipeline architecture are provided in the **Fig. S8.**”
-

Comment #7:

7. Regarding efficiency evaluation and Fig. 6, ADC performance is typically evaluated with more comprehensive metrics beyond area and energy. A detailed comparison table including resolution, sampling frequency, INL, DNL, technology node, etc., would provide a clearer and fairer benchmarking. It is also recommended that the authors extrapolate their design to higher resolutions (8-bit for example) to strengthen the comparison with other state-of-the-art ADCs.

Response:

We agree with the reviewer that a comprehensive comparison table is essential for a fair and insightful benchmark. To address this, we have added a new table (**Table. S6**) to the **Supplementary Information**, providing a detailed comparison of our work with several state-of-the-art ADCs, with a particular focus on those designed for or used in CIM systems.

We also concur that linearity metrics like INL and DNL are crucial. However, these metrics are not uniformly reported across all referenced CIM-focused ADC works. Nevertheless, a common benchmark for high performance is maintaining linearity below 1 LSB. For instance, the excellent work by Khaddam et al.⁵ reports sub-1 LSB performance for their high-resolution ADC. To facilitate future comparisons and demonstrate our design's robustness, we provide a detailed, resolution-dependent analysis of our ADC's simulated INL and DNL under experimentally-validated variation in **Fig. 4 and Table S2**, which confirms strong sub-1 LSB performance within our target operational range.

Regarding the suggestion to extrapolate our design to 8-bit resolution, we respectfully believe a direct comparison at that level would be speculative and detract from the core focus of this study. Our experimental efforts were specifically designed to validate the ADC's performance and advantages in the up to 6-bit range. This is the regime where our adaptive approach offers the most significant system-level benefits for complex neural networks, as demonstrated in our **Fig. 5** analysis. Extrapolating to 8-bits would therefore move beyond the scope of our current experimental validation.

Instead, we have focused on providing a robust comparison within the relevant bitwidth range (up to 6-bit) and against CIM-specific ADC implementations where available.

However, to follow the reviewer's suggestion and conceptually illustrate a path towards higher resolutions, we have now included **Fig. R2.5**. This figure illustrates how our memristor ADC can be used as a sub-converter in a multi-stage pipeline ADC architecture. This approach circumvents the exponential scaling of our flash-like design for high-resolution applications. For example, by cascading four 4-bit sub-ADCs (as shown in **Fig. R2.5B**), a 16-bit converter could be realized. The key advantage is that each sub-ADC only needs to maintain linearity for a low resolution ($n\text{-bit} = 4\text{-bit}$ in this example), making the entire design much more robust to the memristor device variations discussed previously. Furthermore, since each sub-ADC require lower precision, the memristors can operate in a narrower conductance range, leading to potentially lower programming and operating power. This makes the pipeline architecture a very promising direction for scaling our technology to higher resolutions while retaining its core benefits.

Table R2.1. State-of-the-art ADC performance comparison for CIM applications. This table benchmarks the proposed memristor-based ADC against other ADC architectures either specifically designed for or integrated within CIM systems. Performance metrics highlight trade-offs in resolution, linearity, energy, area, and speed across different technology nodes and design approaches. N/A indicates data not available or not applicable; CAM, Content-

Addressable Memory; CCO, Current-Controlled Oscillator; CVH, Current-Voltage-Hybrid; OCCS, Offset-Compensating Current Sensing; SAR, Successive Approximation Register.

Metric	This work	Yang et al. Nature Comm. 2025	Khaddam-Aljameh et al. JSSC 2022	Hsu et al. JSSC 2024	Roy et al. JSSC 2025	Yao et al. Nature 2020	Yin et al. Trans. on Electron Devices 2020	He et al. SSCL 2020	Huo et al. Nat. Electron 2022	Zhang et al. Science 2023
Type	CAM-based	Ramp-based	CCO-based	CVH-based	OCCS + SAR ADC	SAR	Flash	Flash	SAR	Ramp
Resolution (bit)	5	5	12	5	6	8	3	1	8	8
CMOS Node (nm)	16	180	14	22	22	130	90	90	55	130
ADC clk freq. (MHz)	1000	1000	3300	N/A	8.3 or 16.6	N/A	150	140	8	200
Sampling rate (MHz)	1000	31.25	N/A	N/A	N/A	N/A	18.75	17.5	0.015	0.78
Area (μm^2)	24.29	558.03	400	N/A	3180.7	1500	N/A	N/A	N/A	N/A
Power (μW)	12.58	9.3	N/A	N/A	N/A	51	N/A	N/A	33.18	11.9
Energy per Conv	12.58 fJ	42.82 pJ	N/A	N/A	N/A	10 fJ	N/A	N/A	N/A	N/A
Latency per Conv (ns)	0.66	32	N/A	N/A	N/A	10	N/A	N/A	N/A	N/A

Revision:

1. We have added **Table R2.1** as **Table S3** in the revised **Supplementary Information**.
2. We have updated the main manuscript to reference this new comparison. In the “Hardware efficiency evaluation of memristor-based ADC” section, on page 12, the text now states: **“We also designed a full custom layout of the proposed ADC design (Fig. S7). Besides, we conducted a comprehensive analysis comparing our ADC with a detailed comparison against CIM-specific designs provided in Table S3.”**

Comment #8:

8. There are also inconsistencies related to the reported area of the proposed memristor-based ADC. For example, the reported area of $\sim 18 \mu\text{m}^2$ for the 5-bit ADC seems questionable, given that the area of a single memristor device shown in Fig. 3 is on the order of $100 \mu\text{m}^2$. The authors should clarify how the reported area is estimated. In addition, does this estimation includes or excludes peripheral circuitry and whether the reported ADC utilizes super-resolution? A breakdown of the area estimation would be helpful for transparency.

Response:

We appreciate the reviewer's observation regarding the apparent discrepancy in area. The memristor device previously shown in **Fig. 3A** (now replaced by 8×8 memristor crossbar array) is a large-area test structure specifically designed for ease of fabrication and probe station characterization in our university lab environment. It is not representative of the scaled device dimensions used in advanced technology nodes for integrated circuits. The reported area for our 5-bit ADC is an estimation based on a 16nm technology node, leveraging established layout designs from previous analog CAM work. Memristor devices in advanced technology nodes can be scaled down significantly, with active areas often in the nanometer range (e.g., 2nm reported in Pi, S. et al., Nature Nanotechnology 2019)⁶. Therefore, a single memristor device in a 16nm process would occupy a much smaller area than this test structure.

The reported area for our 5-bit ADC ($24.29 \mu\text{m}^2$) is an estimation based on a 16 nm technology node, leveraging established layout designs from previous analog CAM work. To provide tangible validation for this estimation, we have designed a full custom layout of a comparable 5-bit ADC in a UMC 28 nm process, which is now included as **Fig. S7**. This layout, which shows a total area of $43.05 \mu\text{m}^2$, confirms that our area estimates are well-founded and that a compact implementation is feasible. Scaling the Q-cell area from this 28 nm layout to 16 nm further corroborates our estimation.

To enhance transparency, we provide a detailed breakdown of the area estimation in **Fig. 6B** of the main text. This estimation includes the ADC core (which comprises the precharge circuit, source driver, and other essential circuits for the Q-cells) and the decoder circuit. It does not include the memristor write drivers, as these are typically shared resources within a larger CIM system and are not exclusive to the ADC functionality. The super-resolution strategy is implemented at the system level that utilizes two existing ADCs and does not require additional dedicated ADC hardware; it leverages the inherent variations of existing ADCs.

Revision:

1. We have corrected the estimated area of the 5-bit ADC to $24.29 \mu\text{m}^2$ (at 16 nm) throughout the manuscript, including in the **Abstract** and **Discussion**.
2. We have updated **Fig. 6B** in the main manuscript to provide a clear, visual breakdown of this area into its core components.
3. To further validate our estimation methodology, we have added a new figure, **Fig. S7**, to the **Supplementary Information**, which shows a full custom layout of a comparable 5-bit ADC designed in a UMC 28 nm process.
4. In the "Implementation details for 5-bit ADC design" subsection of the Methods, we have clarify the estimation process:

“Our design consists of the ADC core and a decoder circuit, with their respective contributions to energy and area detailed in **Fig. 6B**. The estimations are based on our previous analog CAM in 16 nm technology. To calculate the energy consumption, we began with the baseline analog CAM design, which consumes 0.47 fJ per operation (excluding its DAC portion). To enhance the linearity of our ADC, we expanded the memristor conductance range to 100-400 μS . This primarily affected the source driver within each Q-cell, increasing its energy consumption from 0.29 fJ to 0.58 fJ. Consequently, the total energy for the ADC core is estimated at 12.58 fJ per operation.

For the area estimation, a single Q-cell occupies an area of approximately $0.72 \mu\text{m} \times 0.72 \mu\text{m}$, or $0.52 \mu\text{m}^2$, as shown in the 16-nm physical layout. Therefore, the 5-bit ADC core, comprising 16 Q-cells, has an area of $8.29 \mu\text{m}^2$. The decoder circuit, synthesized using a 28 nm technology library, adds an area of $16.00 \mu\text{m}^2$, resulting in a total estimated ADC area of $24.29 \mu\text{m}^2$. This area estimation for ADC-specific components excludes memristor write drivers since they are treated as shared system-level resources. To further validate this estimation methodology, we designed a full custom layout of a comparable 5-bit ADC in a UMC 28 nm process, which supports our scaled estimations (see **Fig. S7** for details).

The total conversion latency is estimated as 0.165 ns, which combines the core’s 45 ps latency from the CAM reference and the decoder’s 0.12 ns latency from synthesis. This total latency serves as the basis for our system-level timing analysis. While our estimation uses a mixed-technology approach, it represents a conservative projection. A full 16 nm implementation would likely yield further improvements in energy and area. By extrapolating these validated figures, we calculated the total energy and area metrics for various ADC configurations.”

Comment #9:

9. Finally, there are a few minor issues, such as typos or repeated abbreviations in Line 155, Figure S1, and elsewhere. The authors are suggested to carefully proofread the manuscript and supplementary materials for consistency and correctness.

Response:

Thank you for your careful proofreading and for pointing out these important issues. Following your suggestions, we have conducted a comprehensive proofreading of the whole manuscript and Supplementary Information to enhance consistency and correctness.

Revision:

Specifically, we have made the following corrections:

1. We have revised the sentence previously at Line 155 for better clarity and have addressed the typos and related abbreviation definitions in that section: “Given that the fundamental operation of our design relies on precise analog-to-digital conversion, we first evaluated its baseline performance using metrics such as INL and DNL under uniform quantization conditions.”
 2. We have carefully revised **Fig. S1 (now Fig. S6)** and its caption to improve accuracy and clarity.
 - a. In the flowchart, we corrected a grammatical error, changing “Converge achieved?” to the more precise “Convergence achieved?”.
 - b. We also rephrased a key processing step of removing outliers for better readability, changing “Trim sampled data to 0.5% to 99.5%” to “Keep data between the 0.5th and 99.5th percentiles”.
 - c. Furthermore, in the caption, we refined the description of the convergence criteria to be more specific, clarifying that the process continues until “the change in MSE between iterations falls below the threshold.”
 3. In addition, we have performed a comprehensive language polish throughout the manuscript to improve clarity and flow.
 - a. We added “also” to the sentence in Page 12: “Notably, these results not only recover the accuracy loss from device variations but, in some cases (e.g., 4-bit: +4.06% improvement), also surpass the theoretical adaptive quantization baseline.”
 - b. We polished the sentence by moving the word “multiplying” in Page 16: “The step size for the compliance current is determined by multiplying the difference between the measured conductance and the target value (in μS) by an empirical scaling factor.”
 - c. We added “ranging” to the sentence describing target conductance values (Page 17) to make the expression more precise. “Target conductance values are selected ranging from 100 μS to 400 μS in 50 μS steps, with each target programmed 100 times.”
 4. We have systematically standardized the use of abbreviations throughout the paper. For most abbreviations, we have ensured they are defined only once upon their first appearance in the main text to avoid redundancy.
-

Reviewer #3

Response:

We thank Reviewer #3 for their participation in the review process.

Reviewer #4

The paper presents a memristor-based analog-to-digital converter (ADC) architecture designed for in-memory computing (IMC) tiles, with the goal of achieving improved area and energy efficiency. The proposed design utilizes analog content-addressable memory (CAM) cells with programmable overlapped boundaries to establish optimized quantization thresholds. This architectural choice enables favorable integral non-linearity (INL) and differential non-linearity (DNL) performance for mid- and low-precision ADC implementations.

The manuscript includes system-level validation using standard benchmarks such as CIFAR-10 and ImageNet, comparing the proposed solution against existing state-of-the-art designs. The results indicate improvements in both area and energy metrics, which are promising for large-scale deployment of IMC systems.

However, I respectfully find that the overall contribution and technical depth of the work, as currently presented, are limited. While the proposed approach shows incremental advancement, it lacks sufficient novelty and rigorous analysis that would merit publication in a high-impact journal such as *Nature Communications*. The experimental results, though relevant, require further elaboration and comparative evaluation to convincingly demonstrate superiority over existing methods. Below you can find the detailed comments.

Response:

Thank you for the detailed feedback. While we appreciate the acknowledgment that our results are “promising for large-scale deployment of IMC systems”, we have undertaken significant revisions, including new experiments, expanded comparative analyses, and deeper technical discussions, to address the concerns regarding contribution, novelty, and technical depth. We respectfully disagree with the assessment of limited novelty and believe our manuscript, particularly with the extensive new data and analyses, now offers significant advancements that warrant publication in *Nature Communications*.

We have substantially strengthened the manuscript to address each of your primary concerns:

1. **On novelty.** We have revised the introduction to better articulate the novelty, which lies in three key areas. Firstly, this work presents the first demonstration of a memristor-based analog-to-digital converter specifically designed for **hardware-native adaptive quantization** in CIM systems, directly addressing the challenge of diverse data distributions. Secondly, we introduce a novel **super-resolution strategy** that uniquely leverages inherent device variations to enhance quantization precision, in some cases even surpassing the ideal software baseline. Thirdly, the achieved hardware efficiency gains, including a **15.1× improvement in energy and a 12.9× reduction in area** compared to state-of-the-art ADCs, represent a substantial, rather than incremental, advancement for the field.
2. **On rigorous analysis.** To address the need for more rigorous analysis and experimental elaboration, we have significantly expanded our experimental validation. We now present new characterization data from a **fabricated 8×8 memristor crossbar array (64 devices)**. This new evidence, which is now central to our manuscript (**Fig. 3, Figs. S1-S4**), includes: 1) Statistical analysis of array-level **device-to-device (D2D) variation**, which validates our simulation parameters with an experimentally measured standard deviation of only 2.73 μS (<1% relative variation). 2) **Multi-device I-V characterization** and successful **pattern programming**, demonstrating high functional yield and spatial control. 3) **Endurance testing** showing robust performance over 3×10^7

cycles. Furthermore, we have validated our area estimations with a **full custom layout of the 5-bit ADC in a UMC 28 nm process (Fig. S7)** and added a new comprehensive analysis on the ADC's robustness against **dynamic and static voltage variations (Note S3, Fig. S5)**.

3. **On comparative evaluation.** To provide a more convincing demonstration of superiority, we have replaced our previous general comparison with a new, detailed comparison table (**Table S3**) that focuses specifically on **state-of-the-art ADCs designed for or used in CIM systems**. This new table, which includes works cited by the reviewer, benchmarks our design on relevant metrics and clearly highlights its competitive advantages in energy, area, and unique adaptive capabilities within the proper context.

In summary, we believe these extensive revisions, particularly the integration of new experimental array-level data and a full custom layout, have transformed the manuscript by providing the rigorous validation and clear contextualization required for a high-impact publication. We are confident that the work now presents a novel, robust, and thoroughly benchmarked solution to a critical bottleneck in CIM systems.

Revision:

1. We have integrated extensive new experimental data from a fabricated 8×8 memristor array.
 - a. The micrograph of the array and the statistical D2D variation analysis have been made central to the main text as the new **Fig. 3A** and **Fig. 3C**, respectively.
 - b. Additional array characterization, including multi-device I-V curves, pattern programming demonstrations, and endurance data, has been added to the **Supplementary Information** as **Figs. S2, S3, and S4**.
 2. We have added a full custom layout of the 5-bit ADC in a 28 nm process to the **Supplementary Information** as **Fig. S7** to rigorously validate our area estimation methodology.
 3. We have introduced a new comparison table (**Table S3**) to provide a focused benchmark against state-of-the-art ADCs specifically relevant to CIM applications.
 4. We have added a new robustness analysis against voltage variations, with detailed results presented in **Note S3** and **Fig. S5** in the **Supplementary Information**.
 5. We have revised the **Introduction** to better articulate the novelty and significance of our work. On page 2, we clarified the specific challenge we address: **“Moreover, existing quantization methods struggle to handle diverse output distributions in CIM architectures in an energy-efficient manner.”**
-

Comment #1:

On statements and claims in the manuscript:

• "Moreover, existing quantization methods struggle to handle diverse output distributions in CIM architectures". - This may not be correct as many works have dedicated techniques to mitigate these distribution effects. In [Hsu et al. JSSC'23, Khaddam et al. JSSC'22, S. K. Roy et al. JSSC'24], they have introduced several non-ideality mitigation techniques to handle output distributions.

Response:

Thank you for this important point and the valuable references. You are absolutely correct, and we apologize for the imprecise language.

We should clarify that we are addressing a different challenge than the excellent works you cited. These works tackle **hardware non-idealities** through sophisticated techniques: Hsu et al.⁷ employ current-voltage-hybrid readout schemes for signal margin challenges, Khaddam et al.⁵ employ linearized CCO-based ADCs, using digital calibration to address the converter's intrinsic non-linearity and gain/mismatch variations, and Roy et al.⁸ introduce statistical error compensation for wire parasitic effects.

Our work focuses on a complementary problem: **adaptive quantization for diverse output distributions**. This refers to dynamically programming quantization boundaries to match the varying statistical distributions across different network layers and inputs, particularly at low bitwidths, without incurring substantial hardware overhead or complex software-driven calibration loops for the quantization scheme itself.

The cited works make CIM hardware more reliable and accurate through calibration and compensation, while we address the orthogonal challenge of making quantization boundaries directly programmable and adaptive to data characteristics in a hardware-native manner. Our memristor-based approach leverages analog programmability to establish these adaptive boundaries efficiently.

These approaches are complementary rather than competing - non-ideality mitigation and adaptive quantization can work together in practical CIM systems. We have revised the manuscript to clarify this distinction.

Revision:

1. We have revised the **Introduction** section as follows: On page 2, sentence has been revised to: "Moreover, existing quantization methods struggle to handle diverse output distributions in CIM architectures in an energy-efficient manner."
2. We have added a new note, "Note S4: Synergy with array-level compensation techniques for non-idealities," to the **Supplementary Information**. This note provides a detailed analysis of how our ADC architecture complements existing methods for mitigating crossbar array non-idealities.
3. We have added a new paragraph at the end of the **Discussion** in the main manuscript to explicitly reference this detailed discussion. On page 15, we added the following text: "It is important to note that our ADC architecture is designed as a modular component within a larger CIM system. While this work focuses on optimizing the quantization stage, our ADC can operate synergistically with advanced techniques that address circuits-level non-idealities, such as signal margin limitations and IR drop, as well as crossbar device-to-device variations. A detailed discussion on the

orthogonality and complementary nature of our approach with state-of-the-art compensation schemes is provided in **Note S4.**”

Comment #2:

• **"ADCs occupy up to 87.8% of total energy and 75.2% of chip area" - This may not be a general statement and many recent works have presented IMC topologies with much less ADC dominance. A review paper [Hung et al. OJ-SSCS'21] includes such component-wise area and power contributions and those are quite different from what is claimed in this statement.**

Response:

We appreciate the reviewer's critical assessment of this statement and thank them for pointing to the excellent review by Hung et al.⁹ We agree that the proportion of ADC overhead is highly dependent on the specific CIM architecture, technology node, and design choices. Our intention was not to present this figure as a universal truth for all CIM systems.

However, the challenge of ADC dominance is a central and well-documented theme in the CIM field. A recent comprehensive review by Aguirre et al.⁴ (in *Nature Communications*, 2024) reinforces this point, noting that an ADC *"can consume up to 70–90% of the on-chip area ... and up to 80–88% of energy"*. The review concludes that *"ADCs are commonly the largest and most power-hungry circuit block in a memristive neural network"* and that reducing their overhead is *"one of the main challenges in memristor-based ANN hardware design"*. This shows that substantial ADC overhead is a critical and recurring issue, not an isolated observation.

In fact, the review by Hung et al. strongly supports the core argument that ADC and readout peripherals are a primary challenge. The paper highlights that designing for high precision imposes *"readout circuits with large area overhead and high energy consumption"*. More quantitatively, the illustrative example in that review shows that while the memory array itself consumes 18.21% of the energy, the readout system is the dominant contributor: the *"readout circuit"* consumes 25.10% and the *"reference generator"* (which primarily serves the readout) consumes another 26.03%, totaling 51.13% of the energy budget. The paper further concludes that the *"energy consumption during MAC operations was dominated by the IMC computing circuit and readout circuit, which account for 71.51% of the total macro energy"*.

This significant overhead is not an isolated finding and is corroborated by other state-of-the-art works:

Le Gallo et al.¹⁰ (HERMES) reports that while the ADC area is significant (27% of the total, comparable to the PCM array's 36%), its power consumption constitutes approximately 50% of the entire system's budget, far exceeding other components. Khaddam-Aljameh et al.⁵ (HERMES-core) explicitly state: *"One of the main challenges faced during the realization of an IMC system is that the peripheral circuits, especially data converters that interface the crossbar array with the digital world, carry the largest energy overhead and could even dominate the associated latency and area footprint."* They also highlight a key issue with conventional methods: *"voltage-based A/D converters (ADCs) are mostly used... usually employing a large capacitor for integration... This has thus far hampered the realization of large fully-parallel on-chip MVM operations at true $O(1)$ complexity,"* which underscores the value of exploring memristor-based quantization carriers, as we do in our work.

Similarly, the MRAM-based design by Roy et al.⁸ shows that the *"ADC column array, occupying 48.7% of the total area, dominates the core area compared to the 7.9% share of 32-kB... MRAM array."* The paper further generalizes this challenge by stating that as array sizes increase,

“the high-sensitivity readout peripherals in eNVM IMCs today occupy >90% of the total IMC macro area... thereby negating the areal density advantages of eNVM devices.”

The figures we originally cited (87.8% energy and 75.2% area) were from a specific, highly-cited implementation Yao et al.¹¹, intended as an illustrative example of how dominant this bottleneck can become. To avoid any misinterpretation and to provide a more comprehensive context grounded in multiple sources, we have substantially revised the manuscript.

Revision:

We have revised the **Introduction** section to better qualify our statement on ADC overhead. The text now clarifies that these figures represent a notable example rather than a universal constant, reading: *“Current ADCs, as fundamental components for signal conversion, consume excessive energy and circuit area, fundamentally constraining the scalability of CIM systems as neural network complexity grows.”*

Comment #3:

• "Current solutions present a clear trade-off: at low ADC precision (≤ 6 bits), uniform quantization offers hardware-friendly implementation but suffers from significant precision loss" - Many recent works have shown to have promising accuracy results using 5-bit ADCs (for instance, in [Hsu et al. JSSC'23], they use time-interleaved voltage and current comparator based ADCs with changing BL settling voltage conditions to account for changing output distributions.)

Response:

We thank the reviewer for this important clarification and for referencing the excellent work of Hsu et al.⁷. We agree that their work achieves promising accuracy at 5-bit precision, and our revised manuscript now cites it as a key example that underscores our central argument. Our original statement was intended to highlight that achieving such accuracy with low-bit ADCs necessitates moving beyond simple, non-adaptive uniform quantization. The approach in *Hsu et al. JSSC'23* is a prime example of this trade-off. It is not a simple uniform quantizer; rather, it relies on a sophisticated, hardware-state-aware adaptive system to maintain fidelity. Specifically, it employs a DACQ circuit to adaptively regulate the bitline settling voltage based on the degree of computational parallelism (N_{ACCU}) to keep the total current within a manageable range for the ADC. This is a compensatory mechanism for the hardware's operational state, not an adaptation to the data's statistical distribution. **This strategy comes at a potential significant cost.** To manage the bitline current and enable the DACQ, their system processes inputs in sequential blocks, which substantially increases latency. Furthermore, the design requires a suite of tightly co-designed modules, including the DACQ and a complex three-phase hybrid-mode ADC (CVH-ADC), adding to the overall system complexity and overhead.

In contrast, our work proposes a different paradigm of adaptivity. Instead of compensating for fixed hardware parallelism patterns, our memristor-based ADC directly adapts its quantization boundaries to the statistical distribution of the analog outputs. This allows us to minimize quantization error and achieve high accuracy without the latency penalty of sequential block-based processing.

Therefore, we believe our original assertion holds true and is, in fact, strengthened by examples like Hsu et al. It demonstrates that low-bit ADCs require adaptivity to perform well, and different adaptive strategies come with distinct trade-offs in latency, complexity, and power. Our approach offers a highly efficient, one-step conversion pathway to achieve this adaptivity directly in hardware.

Revision:

1. We have revised the **Introduction** section as follows: On page 2, previous sentence has been revised to: "**Current solutions present a clear trade-off: at low ADC precision (≤ 6 bits), uniform quantization generally offers hardware-friendly implementation with low energy and area overhead but suffers from significant precision loss.**"
2. We have added a new note, "Note S4: Synergy with array-level compensation techniques for non-idealities," to the **Supplementary Information**. This note provides a detailed analysis of how our ADC architecture complements existing methods for mitigating crossbar array non-idealities, including a discussion of the work by Hsu et al..

Comment #4:

On the proposed memristor-based ADC solution:

- **Is the proposed design robust to voltage variations, especially the impact of voltage fluctuations on VL and VH seems crucial? And, the impact of these voltage variations across large chip dimensions could be quite critical to analyse. The suggestion is to investigate into these impacts and report them.**

Response:

Thank you for this insightful question regarding voltage variation robustness. We agree this is critical for practical implementation and have conducted comprehensive analysis to address this concern.

We performed parametric analysis across ADC resolutions (2-5 bit) under two types of voltage variations on VH and VL: dynamic power supply noise (0.05-20 mV peak-to-peak) modeling rapid fluctuations, and static DC drift (0-50 mV) representing slow voltage shifts across the chip. Results are based on 10 Monte Carlo runs per parameter point, with details in **Note S3** and **Fig. S5 (shown below as Fig. R4.1)**.

The ADC shows excellent resilience to dynamic variations. For a 5-bit ADC, increasing noise from 0.05 mV to 10 mV only degrades INL/DNL from 0.306/0.423 to 0.344/0.485 LSB. Even at substantial 20 mV noise, performance remains acceptable with INL/DNL of 0.464/0.521 LSB. This demonstrates that the ADC can effectively withstand significant dynamic noise up to 10 mV. While nonlinearity increases to 20 mV noise, performance remains at low LSB, especially for 2-4 bit resolutions where the impact is minimal across the entire tested noise range.

Static DC offset on VH and VL has a more pronounced impact on linearity. To quantify this, we selected 10 mV dynamic noise levels as representative cases. Our analysis of the 5-bit ADC shows that when the static DC offset increases from 0 mV to 50 mV, performance degrades: At 10 mV noise, the INL increases from 0.344 LSB to 0.523 LSB, and the DNL increases from 0.485 LSB to 0.731 LSB. However, this highlights our design's core advantage: unlike conventional ADCs requiring complex peripheral compensation, our architecture enables in-situ recalibration by updating the conductance-boundary map. This directly adjusts quantization thresholds to compensate for drifted VH/VL levels without additional hardware overhead.

For applications where frequent recalibration is not feasible, maintaining DC drift within ~25 mV ensures sub-0.4 LSB INL for 5-bit configuration. This analysis demonstrates that our ADC is robust to dynamic noise and features a unique hardware-efficient mechanism to nullify static drift, ensuring reliable performance in realistic chip environments.

Fig. R4.1. Simulated ADC linearity robustness against reference voltage variations. The plots show the RMS of the maximum INL (left column) and DNL (right column) for ADC resolutions from 2-bit to 5-bit. The analysis evaluates the impact of dynamic power noise (x-axis, ranging from 0.05 to 20 mV p-p) under three distinct static DC drift conditions (0, 25, and 50 mV) on the reference voltages V_H and V_L . Each data point represents the RMS of the maximum values gathered from 10 Monte Carlo simulation runs, ensuring a robust assessment of performance under a wide range of realistic operating conditions.

Revision:

1. We have added **Fig. R4.1** as **Fig. S5** in the revised version.
2. We have added a new sentence to the end of the “Experimental benchmarking of uniform quantization ADC” section: “Beyond device conductance variations, the ADC also demonstrates strong resilience to dynamic power supply noise and offers an in-situ recalibration mechanism to counteract static DC drifts, as detailed in **Note S3** and **Fig. S5**.”

3. We have added **Note S3**, “**Robustness analysis against reference voltage variations**

To evaluate the robustness of the proposed ADC against practical voltage instabilities, we performed a comprehensive parametric analysis. The study focused on the impact of two primary non-idealities on the ADC’s linearity metrics (INL and DNL): dynamic power supply noise and static DC drift, applied to the reference voltages V_H and V_L . The simulation was structured across a three-dimensional parameter space under experimental device variation, with results plotted in **Fig. S5**. To ensure statistical significance, each parameter point was evaluated through 10 Monte Carlo simulation runs, with 1% random variation applied to memristor conductance.

S3.1 Impact of dynamic power noise

Our analysis evaluates the ADC’s intrinsic robustness to high-frequency dynamic power noise, which represents random fluctuations that cannot be compensated for by recalibration. The plots in first row of **Fig. S5** (Voltage Drift = 0.0 mV) specifically illustrate this resilience.

For the 5-bit ADC under zero DC drift, the architecture maintains excellent linearity even under significant noise. Increasing noise from 0.05 mV to 10 mV results in only minor degradation of linearity, with the RMS of maximum DNL increasing from 0.423 LSB to 0.485 LSB, and INL increasing from 0.306 LSB to 0.344 LSB. Even at a high noise level of 20 mV, the ADC maintains robust performance, with DNL and INL remaining at 0.521 LSB and 0.464 LSB, respectively. This demonstrates the architecture’s inherent stability in noisy mixed-signal environments, which is a crucial characteristic for reliable operation in practical CIM systems.

S3.2 Impact of static DC drift

To quantify the impact of static DC drift, we analyzed ADC performance under two noise conditions representing low (1 mV) and moderate (10 mV) system environments. When the static DC offset increases from 0 mV to 50 mV, the results show different sensitivity levels depending on the background noise.

In the low-noise environment (1 mV), performance degradation is modest: maximum DNL increases from 0.565 to 0.604 LSB, while maximum INL rises from 0.375 to 0.417 LSB. However, under moderate noise conditions (10 mV), the impact becomes more pronounced, with DNL increasing from 0.485 to 0.731 LSB and INL from 0.344 to 0.523 LSB.

However, this highlights our design’s core advantage: unlike conventional ADCs requiring complex peripheral compensation, our architecture enables in-situ recalibration by updating the conductance-boundary map. This directly adjusts quantization thresholds to compensate for drifted V_H/V_L levels without additional hardware overhead.”

4. We have added the new subsection to **Methods** for better clarity and focus. On page 21, the subsection title and text now read:

“Robustness to voltage variations and in-situ recalibration

To assess the robustness of our ADC in practical chip environments, we conducted comprehensive simulation-based analysis of its resilience to reference voltage variations. Our evaluation systematically investigated the impact of both high-frequency dynamic power supply noise and quasi-static DC voltage drift on ADC linearity (INL/DNL). The results show that the ADC is highly resilient to dynamic noise, as demonstrated by the 5-bit ADC which maintains INL well below 0.5 LSB even under 20 mV peak-to-peak noise.

While the analysis indicates higher sensitivity to static DC drift, this highlights a core strength of our architecture. Unlike conventional designs that require dedicated compensation circuits, our

ADC can directly counteract such drifts through in-situ recalibration. Since memristor conductances define the quantization boundaries, any quasi-static voltage drift can be completely nullified by reprogramming the memristors to new conductance values corresponding to the desired thresholds under the drifted condition. This ability to directly reconfigure quantization boundaries eliminates the need for additional hardware, thus demonstrating significant advantages in efficiency and design simplicity (detailed methodology and results in **Note S3** and **Fig. S5**).

Comment #5:

• Typically, non-idealities dominated by device-to-device variations in the memory array and input-dependent IR drop on the access lines severely impact the accuracy of the IMC tiles. This work does not necessarily address those idealities. There are some recent works that tackle these issues on a chip level [Khaddam et al. JSSC'22, S. K. Roy et al. JSSC'24]. The suggestion is to investigate into these impacts and report them.

Response:

We thank the reviewer for raising these important system-level considerations. We acknowledge that D2D variations in the main CIM crossbar array and input-dependent IR drop on access lines are critical factors affecting overall CIM accuracy. The cited works^{5,8} represent the state-of-the-art in addressing these crucial challenges.

We position our ADC as a modular and essential component within the broader CIM pipeline, which is designed to work in synergy with other techniques that address non-idealities at the array level, rather than replace them.

The excellent work by [S. K. Roy et al. JSSC'24]⁸ introduces statistical error compensation (SEC), a powerful system-level algorithmic technique that pre-compensates for non-linearity arising from wire parasitics (i.e., IR drop) by scaling the inputs. Our adaptive ADC is perfectly suited to quantize the resulting, more linear analog signal generated by the VMM array. The SEC algorithm ensures the integrity of the analog dot product, while our ADC ensures this result is digitized with high efficiency and accuracy. This demonstrates a clear path for combining these orthogonal but complementary approaches to build a highly robust CIM system.

Similarly, the HERMES-core presented by [Khaddam et al. JSSC'22]⁵ employs a feed-forward compensation technique to linearize its CCO-based ADC's transfer curve, partially addressing the read voltage drop seen by the ADC itself. While this is an effective solution for that specific ADC topology, our architecture's strength lies in its in-situ recalibration capability. The quantization boundaries in our design are defined by programmable memristor conductances. This offers a potential pathway to compensate not only for local reference voltage drifts but also for chip-scale variations, such as transistor mismatch or quasi-static voltage shifts induced by IR drop, by reprogramming the memristors. This adaptability is intrinsic to our design and does not require complex peripheral compensation circuits.

Therefore, our contribution is orthogonal yet complementary to these crucial array-level solutions. A comprehensive, high-performance CIM system would ideally integrate a suite of techniques, including array-level compensation and an efficient, adaptive quantization engine like our memristor-based ADC. While a full experimental characterization of our ADC integrated with these specific system-level compensation schemes is an important direction for future work, it is beyond the scope of the current manuscript, which focuses on establishing the performance and viability of the novel

ADC architecture itself. To better articulate this positioning, we have added a dedicated note in the **Supplementary Information (Note S4)** to discuss this system-level context in detail.

Revision:

1. We have added a new note, “**Note S4: Synergy with array-level compensation techniques for non-idealities,**” to the **Supplementary Information**. This note provides a detailed analysis of how our ADC architecture complements existing methods for mitigating crossbar array non-idealities.
2. On page 15-16, we added the following text: “**It is important to note that our ADC architecture is designed as a modular component within a larger CIM system. While this work focuses on optimizing the quantization stage, our ADC can operate synergistically with advanced techniques that address array-level non-idealities, such as signal margin limitations, IR drop and crossbar device-to-device variations. A detailed discussion on the orthogonality and complementary nature of our approach with state-of-the-art compensation schemes is provided in Note S4.**”

Comment #6:

- **Are the ADC related circuits such as memristor write drivers and control logic included in the ADC area calculations? The suggestion is to include the breakdown of the 18µmsq ADC area number (a layout photo perhaps) and report it.**

Response:

We appreciate the request for clarification on the ADC area calculations. In our revised manuscript, we have addressed this in detail and corrected a previous miscalculation. Our updated area for the 5-bit ADC is **24.29 µm²** (estimated at 16 nm). The original manuscript mistakenly reported a slightly smaller area due to a misinterpretation of a reference layout, for which we apologize. We have thoroughly re-evaluated this. The updated area breakdown, shown in the revised **Fig. 6B** (shown here as **Fig. R4.2**), is as follows: ADC Core (16 Q-cells): 8.29 µm² (16 × 0.52 µm²/Q-cell). Decoder Circuit: 16.00 µm².

Fig. R4.2. Estimated energy and area comparison of memristor-based ADC. (A) Scatter plot comparing energy efficiency versus area of the state-of-the-art ADCs¹², with technology nodes indicated by the color bar. Our memristor-based ADC (red star) demonstrates superior performance in both metrics, compared to conventional designs, including lowest-energy design¹³ (blue triangle) and smallest-area design¹⁴ (green square). (B) Breakdown of energy consumption and area utilization of our ADC implemented at 5-bit precision, showing the proportion between ADC core (precharge circuit, source driver, and other circuits) and decoder.

To provide tangible evidence for our area estimation, we have designed a **full custom layout** of the 5-bit ADC in UMC 28 nm process (**Fig. R4.3**). The layout shows the ADC core ($34.39 \mu\text{m}^2$) and the program driver ($8.67 \mu\text{m}^2$). Scaling the 28 nm Q-cell area ($2.15 \mu\text{m}^2$) to 16 nm yields $\sim 0.70 \mu\text{m}^2$, which strongly validates our corrected reference-based estimation of $0.52 \mu\text{m}^2$.

Regarding the scope of the calculation, the area includes the core Q-cells and the essential decoder logic. It excludes the memristor write drivers for the following reason: in a complete CIM system, the write drivers (programming DACs/ADCs) are a shared resource used for programming both the main VMM array weights and the ADC quantization boundaries. Since boundary programming is infrequent, attributing the entire area of this shared infrastructure to the ADC alone would inaccurately inflate its specific overhead.

Fig. R4.3. Layout of proposed 5-bit memristor-based ADC in a UMC 28 nm process. This layout, created to validate our area estimation methodology, consists of 16 Q-cells arranged in a 4×4 array and an associated program driver. The total area of the design is $43.05 \mu\text{m}^2$, with the Q-cell array occupying $34.39 \mu\text{m}^2$ and the program driver occupying $8.67 \mu\text{m}^2$.

Revision:

1. We have added a new figure, **Fig. S7**, in the revised **Supplementary Information**, showing the full custom layout of our 5-bit ADC in a UMC 28 nm technology with a detailed area breakdown.
2. We have updated **Fig. 6B** in the main text to show the updated area breakdown of the 5-bit ADC, estimated at $24.29 \mu\text{m}^2$.

Comment #7:

• **The dimensions of the memristor device seems quite large, in the order of 10 μ m range (Fig. 3A). However, the area of the ADC using several such devices seems quite small (only 18 μ m²) as claimed in the paper. A more detailed area calculations may be helpful for the readers.**

Response:

Thank you for this crucial observation. You are right to question this apparent inconsistency between the large device in **Fig. 3A** and our small ADC area estimate.

The micrograph shows a large-area test structure fabricated in our university clean room, with dimensions intentionally enlarged to facilitate reliable electrical probing with standard lab equipment. This is not representative of scaled devices used in integrated circuits at advanced technology nodes.

For our area estimation, we assume memristor devices monolithically co-integrated with the CMOS process, where the chip footprint is determined by the underlying CMOS access transistors within the Q-cell. Our detailed area breakdown shows this clearly. For the UMC 28nm process used in our custom layout validation (**Fig. R4.3**), the 2T2R cell occupies a footprint of 430nm \times 250nm (0.11 μ m²), while the memristor itself measures only 130nm \times 165nm (0.021 μ m²) in the stacked layers above. The area is dominated by CMOS components, not the memristors.

Our 24.29 μ m² estimation for the 5-bit ADC assumes a Q-cell area of 0.52 μ m² in 16nm technology, derived from established analog CAM layouts. To validate this projection, our full custom layout in UMC 28nm (**Fig. R4.3**) occupies 43.05 μ m², demonstrating that compact integration is feasible and our 16nm projections are realistic.

Revision:

1. We have added a full custom layout of the 5-bit ADC designed in a UMC 28 nm process as **Fig. S7** in the revised **Supplementary Information**. This demonstrates the feasibility of achieving a compact area in a scaled technology node.
2. We have expanded the **Methods** section (Implementation details for 5-bit ADC design) to detail the area estimation methodology.

Comment #8:

On the results:

• **Fig. 6A compares with non-IMC ADCs (for instance, high-speed IO ADC in [48] is signed 8-bit and is not designed for IMC tiles). Plus, [46] ADC review does not include IMC related ADC implementations, rather they target high-speed IO, DSP applications etc., and they are typically mid- to high A/D resolution implementations.]. The suggestion is to include IMC related ADC designs for a relevant comparison.**

Response:

We appreciate this suggestion to refine our comparison. The reviewer is correct that **Fig. 6A** (and the referenced Murmann survey) includes a broad range of ADCs, many of which are not specifically designed for CIM. Our initial intent was to show our ADC’s standing against the general state-of-the-art in terms of energy and area efficiency for low-to-mid resolution ADCs.

To provide a more focused and relevant comparison for the CIM context, we have revised our benchmarking. The new **Table R4.1** now specifically includes and highlights more ADCs designed for or used in CIM applications. This table features works such as Yang et al. (NL-ADC), Khaddam et al. (CCO-ADC), Hsu et al. (CVH-ADC), Roy et al. (MRAM OCCS), and Yao et al. (SAR). This focused benchmark provides a clearer picture of our ADC’s advantages and trade-offs within the CIM domain, highlighting its state-of-the-art energy and area efficiency combined with its unique hardware-adaptive capabilities.

Table R4.1. State-of-the-art ADC performance comparison for CIM applications. This table benchmarks the proposed memristor-based ADC against other ADC architectures either specifically designed for or integrated within CIM systems. Performance metrics highlight trade-offs in resolution, linearity, energy, area, and speed across different technology nodes and design approaches. N/A indicates data not available or not applicable; CAM, Content-Addressable Memory; CCO, Current-Controlled Oscillator; CVH, Current-Voltage-Hybrid; OCCS, Offset-Compensating Current Sensing; SAR, Successive Approximation Register.

Metric	This work	Yang et al. Nature Comm. 2025	Khaddam-Aljameh et al. JSSC 2022	Hsu et al. JSSC 2024	Roy et al. JSSC 2025	Yao et al. Nature 2020	Yin et al. Trans. on Electron Devices 2020	He et al. SSCL 2020	Huo et al. Nat. Electron 2022	Zhang et al. Science 2023
Type	CAM-based	Ramp-based	CCO-based	CVH-based	OCCS + SAR ADC	SAR	Flash	Flash	SAR	Ramp
Resolution (bit)	5	5	12	5	6	8	3	1	8	8
CMOS Node (nm)	16	180	14	22	22	130	90	90	55	130
ADC clk freq. (MHz)	1000	1000	3300	N/A	8.3 or 16.6	N/A	150	140	8	200
Sampling rate (MHz)	1000	31.25	N/A	N/A	N/A	N/A	18.75	17.5	0.015	0.78
Area (μm^2)	24.29	558.03	400	N/A	3180.7	1500	N/A	N/A	N/A	N/A
Power (μW)	12.58	9.3	N/A	N/A	N/A	51	N/A	N/A	33.18	11.9
Energy per Conv	12.58 fJ	42.82 pJ	N/A	N/A	N/A	10 fJ	N/A	N/A	N/A	N/A
Latency per Conv (ns)	0.66	32	N/A	N/A	N/A	10	N/A	N/A	N/A	N/A

Revision:

1. We have added the new CIM-focused comparison table as **Table S3** in the revised **Supplementary Information**.
-

Comment #9:

• If we compare with IMC ADC, An 8-bit ADC implementation has <1 LSB INL and DNL with area around 400umsq [Khaddam et al. JSSC'22]. But this also includes all kinds of calibration and non-idealities compensation related circuits. These kind of works must be included in the manuscript for comparison.

Response:

Thank you for this important reference. We have included [Khaddam et al. JSSC'22]⁵ (HERMES-core) in our benchmark comparison table (**Table R4.1**) and discuss it in the manuscript.

The HERMES-core achieves <1 LSB INL/DNL at 12-bit resolution with 400 μm^2 area in 14nm, using linearized CCO-based ADCs and sophisticated calibration circuits. Our 5-bit memristor ADC occupies 24.29 μm^2 (estimated at 16nm) with simulated 0.319/0.419 LSB INL/DNL.

While we acknowledge the resolution difference, this comparison illustrates complementary design philosophies. HERMES targets high precision through digital calibration overhead. Our approach prioritizes compactness (16.5 \times smaller area) and hardware-native adaptive quantization for low-to-mid bit applications. The programmable memristors enable direct quantization boundary adaptation without peripheral calibration loops.

We believe these approaches address different needs in the CIM design space—high-precision applications versus area/energy-constrained scenarios requiring adaptive quantization. Importantly, our memristor-based approach inherently provides adaptivity without requiring the extensive calibration infrastructure that traditional ADCs need for comparable functionality.

Revision:

1. We have added a comparison with Khaddam et al. in the new benchmark table (**Table S3**) in the **Supplementary Information**.

Comment #10:

• For the results shown in Fig. 4, the trend with increasing resolution seems problematic. 1-1.5% device variations already seems quite optimistic, especially if the number of such ADCs are large and still the INLs/DNLs are around 2.

References (these are, in my opinion, only a selected few to support my comments.):

[Hsu et al. JSSC'23] Hsu, Hung-Hsi, et al. "A nonvolatile AI-edge processor with SLC–MLC hybrid ReRAM compute-in-memory macro using current–voltage-hybrid readout scheme." *IEEE Journal of Solid-State Circuits* 59.1 (2023): 116-127.

[Khaddam et al. JSSC'22] Khaddam-Aljameh, Riduan, et al. "HERMES-core—A 1.59-TOPS/mm² PCM on 14-nm CMOS in-memory compute core using 300-ps/LSB linearized CCO-based ADCs." *IEEE Journal of Solid-State Circuits* 57.4 (2022): 1027-1038.

[S. K. Roy et al. JSSC'24] S. K. Roy et al., "Compute SNDR-Boosted 22-nm MRAM-Based In-Memory Computing Macro Using Statistical Error Compensation," in *IEEE Journal of Solid-State Circuits*

[Hung et al. OJ-SSCS'21] Hung, Je-Min, et al. "Challenges and trends of nonvolatile in-memory-computation circuits for AI edge devices." *IEEE Open Journal of the Solid-State Circuits Society* 1 (2021): 171-183.

Abhairaj Singh

IBM Research Zurich

Response:

Thank you for your careful examination of Fig. 4. We appreciate the opportunity to clarify the performance of our ADC architecture in light of our experimental device data.

Our new characterization of an 8×8 memristor array (64 devices) shows a standard deviation in programmed conductance of only 2.73 μS over a 100 μS to 400 μS range (see Fig. R4.5). This corresponds to a D2D variation of approximately 0.9% (e.g., 2.73 μS / 300 μS), which addresses your concern about performance in larger arrays. Therefore, the 1% variation used in our Fig. 4 simulations is a conservative, experimentally-grounded parameter, not an optimistic one. This level of device noise is also consistent with state-of-the-art memristor arrays^{15,16}.

The parametric study in Figs. 4C-D is designed to explore the ADC's sensitivity to a wide range of device variations (0% to 2.5%). We must clarify a key point: the high INL/DNL values of ~2 LSB you noted occur only at a hypothetical variation of 2.5% for a 6-bit ADC. In contrast, at our experimentally-validated 1% variation level, the performance is excellent. Our simulations show that a 6-bit ADC maintains sub-1-LSB linearity (maximum INL ≈ 0.72 LSB and DNL ≈ 0.94 LSB).

We present the 6-bit case to demonstrate the performance boundary of a single ADC, which highlights the value of our proposed super-resolution strategy. This approach allows us to achieve even higher effective precision by combining robust, lower-resolution ADCs, thus bypassing the challenge of pushing a single ADC to its physical limits.

To make this connection clearer in the manuscript, we have updated Fig. 4 with a vertical marker at the 1% variation level and revised the text to explicitly link our strong experimental data to the simulated ADC performance (Fig. R4.8).

Fig. R4.4. Optical micrographs of the fabricated 8×8 memristor array (64 devices). The devices utilize a Pt/TaO_x/Ta material stack on Si/SiO₂ substrate. Images show (left) the complete array with contact pads, (middle) magnified view of the crossbar structure, and (right) individual device cells. TE: top electrode; BE: bottom electrode.

Fig. R4.5. Experimental device-to-device (D2D) variation measured across all 64 devices of the 8×8 array. (A) The standard deviation (σ) of the programmed conductance is shown for seven distinct target states. The variation remains consistently stable, with σ values around 2.73 μ S. **(B)** Conductance distributions for the seven target states (100, 150, 200, 250, 300, 350, and 400 μ S), achieved using a program-and-verify scheme. The distinct, well-separated peaks confirm that multiple states can be reliably programmed across the array.

Fig. R4.6. I-V switching characteristics of memristor devices from the 8×8 array. Ten consecutive voltage sweep cycles are shown for five devices at different array locations, demonstrating consistent bipolar switching behavior and uniform performance across the fabricated array.

Fig. R4.7. Demonstration of multi-state conductance programming on the 8×8 memristor array. Seven different letter patterns (‘H’, ‘K’, ‘U’, ‘Re’, ‘A’, ‘D’, ‘C’) were successfully programmed across the array with the color map representing measured conductance values. The clear formation of all patterns confirms high device yield and precise conductance state control across individual devices.

Fig. R4.8. Simulated performance metrics of the memristor-based ADC. (A) INL of 4-bit ADC under 1% variation across 16 digital codes, maximum 0.215 LSB. (B) DNL under the same conditions, maximum 0.322 LSB. (C-D) RMS analysis of maximum INL (C) and DNL (D) under various device variations for ADC resolutions from 2 to 6 bits, illustrating the ADC’s sensitivity to such variations.

Revision:

1. We have updated **Figs. 4C and 4D** in the main manuscript by adding a vertical dashed line at the ~1% variation level. This visually anchors the simulation-based parametric sweep to our experimentally measured device-to-device variation (presented in **Fig. 3C**), highlighting the ADC’s robust performance in the relevant operational region.
2. We have revised the caption for **Fig. 4** to explicitly clarify that panels C and D represent a parametric study. The caption now includes: “(C-D) RMS analysis of maximum INL (C) and DNL (D) under various device variations for ADC resolutions from 2 to 6 bits, illustrating the ADC’s sensitivity to such variations.”
3. We have strengthened the discussion in the “Experimental benchmarking of uniform quantization ADC” section (Page 7) to explicitly link the experimental variation data to the simulation parameters, stating: “These experimental characterizations (including **Fig. S1-S3**) validate the variation parameters to be used in our following simulation evaluations.”

Reference

1. Self-rectifying bipolar TaO_x/TiO₂ RRAM with superior endurance over 10¹² cycles for 3D high-density storage-class memory. <https://ieeexplore.ieee.org/document/6576643/authors#authors>.
2. Zhang, J. *et al.* A 28nm 4Mb Embedded RRAM IP with Record-High Endurance of 10⁷ Cycles and 10 Years@125°C Retention through Reliability-Enhanced Design-Technology Co-Optimization. in *2024 IEEE International Electron Devices Meeting (IEDM) 1–4* (IEEE, San Francisco, CA, USA, 2024). doi:10.1109/iedm50854.2024.10873311.
3. Zeissler, K. A RRAM that endures. *Nat Electron* **7**, 1065–1065 (2024).
4. Aguirre, F. *et al.* Hardware implementation of memristor-based artificial neural networks. *Nat Commun* **15**, 1974 (2024).
5. Büchel, J. *et al.* Efficient scaling of large language models with mixture of experts and 3D analog in-memory computing. *Nat Comput Sci* **5**, 13–26 (2025).
6. Khaddam-Aljameh, R. *et al.* HERMES-Core—A 1.59-TOPS/mm² PCM on 14-nm CMOS In-Memory Compute Core Using 300-ps/LSB Linearized CCO-Based ADCs. *IEEE J. Solid-State Circuits* **57**, 1027–1038 (2022).
7. Pi, S. *et al.* Memristor crossbar arrays with 6-nm half-pitch and 2-nm critical dimension. *Nature Nanotech* **14**, 35–39 (2019).
8. Hsu, H.-H. *et al.* A Nonvolatile AI-Edge Processor With SLC–MLC Hybrid ReRAM Compute-in-Memory Macro Using Current–Voltage-Hybrid Readout Scheme. *IEEE J. Solid-State Circuits* **59**, 116–127 (2024).
9. Roy, S. K. *et al.* Compute SNDR-Boosted 22-nm MRAM-Based In-Memory Computing Macro Using Statistical Error Compensation. *IEEE J. Solid-State Circuits* **60**, 1092–1102 (2025).
10. Hung, J.-M., Jhang, C.-J., Wu, P.-C., Chiu, Y.-C. & Chang, M.-F. Challenges and Trends of Nonvolatile In-Memory-Computation Circuits for AI Edge Devices. *IEEE Open J. Solid-State Circuits Soc.* **1**, 171–183 (2021).
11. Le Gallo, M. *et al.* A 64-core mixed-signal in-memory compute chip based on phase-change memory for deep neural network inference. *Nat Electron* 1–14 (2023) doi:10.1038/s41928-023-01010-1.
12. Yao, P. *et al.* Fully hardware-implemented memristor convolutional neural network. *Nature* **577**,

641–646 (2020).

13. Boris, M. ADC Performance Survey 1997-2024. *GitHub* [Online]. Available: <https://github.com/bmurmman/ADC-survey>.
14. Bindra, H. S., Annema, A.-J., Louwsma, S. M. & Nauta, B. A 0.2 - 8 MS/s 10b flexible SAR ADC achieving 0.35 - 2.5 fJ/conv-step and using self-quenched dynamic bias comparator. in *2019 Symposium on VLSI Circuits C74–C75* (2019). doi:10.23919/VLSIC.2019.8778093.
15. Yonar, A. S. *et al.* An 8b 1.0-to-1.25GS/s 0.7-to-0.8V Single-Stage Time-Based Gated-Ring-Oscillator ADC with 2\times Interpolating Sense-Amplifier-Latches. in *2023 IEEE International Solid-State Circuits Conference (ISSCC)* 1–3 (2023). doi:10.1109/ISSCC42615.2023.10067745.
16. Rao, M. *et al.* Thousands of conductance levels in memristors integrated on CMOS. *Nature* **615**, 823–829 (2023).
17. Jiang, M., Shan, K., He, C. & Li, C. Efficient combinatorial optimization by quantum-inspired parallel annealing in analogue memristor crossbar. *Nat Commun* **14**, 5927 (2023).

Response Letter to Reviewers' Comments

We are very grateful to the reviewers for their time and invaluable feedback throughout the review process. The positive reception to our revisions is encouraging, and the final comments have been helpful in further refining the manuscript's presentation. Our point-by-point responses are provided below.

Reviewer #1

The authors have made a good revision of the manuscript. The new part of the endurance is useful. I think the paper should be accepted.

Still, the manuscript could be further improved indicating:

- 1 – In how many devices has endurance been tested and what is the endurance achieved in each of them.**
- 2 – If endurance has been tested in more than one device, may be they could include other plots in Figure S4.**
- 3 – It would be even better if, for at least one device, the authors show the Voltage/Current versus time plot (measured with high temporal resolution) to observe how the devices switch and make sure that the resistance is stable. Showing read, write, read, erase, read for 5-10 cycles should be sufficient.**

I think the authors may provide this Information if they can. But I don't need to review what they answer again, from my side this is optional (although highly recommended).

Congratulations to the authors for their excellent results.

Response:

We sincerely thank the reviewer for the strong support and these excellent suggestions. We agree that the proposed experiments would add valuable details.

Regarding endurance testing (Points 1 and 2), endurance testing involves time-intensive cycling until complete device failure. Our test yielded $>3 \times 10^7$ cycles on a representative device (**Supplementary Fig. 4**). While this comprehensive test focused on one device, Set/Reset operations have been successfully performed across multiple devices for various measurements. The low device-to-device variation ($\sigma = 2.73 \mu\text{S}$) across all 64 devices (**Fig. 3c**) supports the broader applicability of this endurance performance.

For the switching characteristics (Point 3), while we don't have dedicated time-domain plots, the I-V curves in **Supplementary Fig. 2** were captured through controlled voltage sweeps over time, inherently reflecting temporal switching behavior across 10 consecutive cycles on each device. Combined with the stability data in **Fig. 3d** over 1,000 read cycles, these measurements address the key switching and retention characteristics.

We deeply appreciate your expert guidance throughout this process.

Reviewer #2

The manuscript has been significantly improved and most of my previous concerns have been resolved. I don't have further questions.

Response:

We sincerely thank Reviewer #2 for the positive feedback and for noting the significant improvements in the manuscript. We are grateful for their constructive guidance throughout the review process, which has been invaluable in improving our work.

Reviewer #3

Response:

We thank Reviewer #3 for their participation in the co-review process and their valuable contribution to the manuscript evaluation. We appreciate the thorough and constructive feedback provided through this collaborative review approach.

Reviewer #4

I sincerely thank the authors for their time and effort in preparing the revised manuscript and the response letter in response to my comments. The authors have addressed all my comments.

Response:

We sincerely thank Reviewer #4 for their thorough review and positive acknowledgment. Their comprehensive feedback was invaluable in strengthening the manuscript, and we are pleased that our revisions have fully addressed their comments.